# A²FM: An Adaptive Agent Foundation Model for Tool-Aware Hybrid Reasoning

**Qianben Chen**[1,*] **Jingyi Cao**[1,2,*] **Jiayu Zhang**[1,3,*] **Tianrui Qin**[1], **Xiaowan Li**[1,4], **King Zhu**[1],
**Dingfeng Shi**[1], **He Zhu**[1], **Minghao Liu**[5], **Xiaobo Liang**[6], **Ge Zhang**[7], **Jian Yang**[8],
**Yuchen Eleanor Jiang**[1], **Wangchunshu Zhou**[1,†]

[1] OPPO AI Agent Team    [2] The Chinese University of Hong Kong, Shenzhen
[3] The Chinese University of Hong Kong    [4] University of Science and Technology of China
[5] 2077AI    [6] Soochow University    [7] ByteDance    [8] Beihang University
chenqianben1997@gmail.com, wcszhou@outlook.com

## Abstract

Large language models split into two families: reasoning-centric LLMs, which strengthen internal chain-of-thought reasoning but cannot invoke external tools, and agentic LLMs, which learn to interact with environments and leverage tools but often lag in deep reasoning. This divide arises from fundamentally different training objectives, leading to mismatched strengths and inefficiency on simple queries, where both families tend to overthink or over-call tools. In this work, we present Adaptive Agent Foundation Model (A²FM), a unified framework that follows a route-then-align principle: the model first learns task-aware routing and then aligns mode-specific trajectories under a shared backbone. To address the inefficiency gap, we introduce a third instant mode that handles simple queries directly, preventing unnecessary reasoning or tool calls while complementing the agentic and reasoning modes. To jointly enhance accuracy and efficiency, we propose Adaptive Policy Optimization (APO), which enforces adaptive sampling across modes and applies a cost-regularized reward. On the 32B scale, A²FM achieves 13.4% on BrowseComp, 70.4% on AIME25, and 16.7% on HLE, setting new SOTA among comparable models and performing competitively with frontier LLMs across agentic, reasoning, and general benchmarks. Notably, the adaptive execution achieves a cost of pass of only $0.00487 per correct answer—cutting cost by 45.2% relative to reasoning and 33.5% relative to agentic, thus delivering substantially higher cost efficiency while maintaining comparable accuracy.

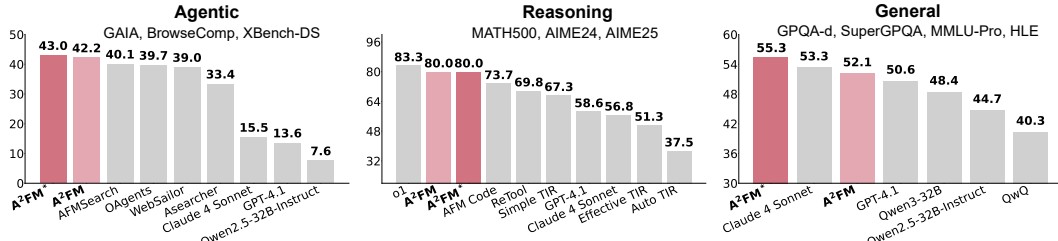

Figure 1: Average performance on agentic, reasoning, and general (ARG) benchmarks. Overall, A²FM* ranks 1st, 2nd, and 1st on the three categories, respectively. Moreover, A²FM*, denoted as a variant that uses the best-suited mode for each benchmark, further improves over the adaptive version by +0.8 on agentic and +3.2 on general benchmarks.

---

*Equal contribution
†Corresponding author

# 1 INTRODUCTION

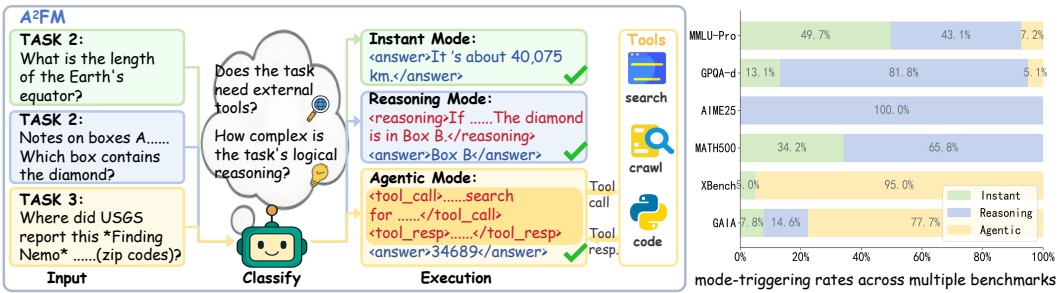

Figure 2: Overview of A²FM. Left: the framework integrates three execution modes—*instant*, *reasoning*, and *agentic*—under a unified backbone with task-aware routing. Right: mode allocation across six benchmarks (MMLU-Pro, GPQA-d, AIME25, MATH500, Xbench-DeepSearch (Xbench), and GAIA-text (GAIA), illustrating how A²FM adapts routing to task characteristics.

Large language models have evolved along two largely divergent directions. Reasoning-centric models such as OpenAI o3 (OpenAI, 2025c) and DeepSeek-R1 (Guo et al., 2025) focus on strengthening internal chain-of-thought reasoning for mathematics, scientific diagrams, and logical tasks. Their training pipelines rely on text-only corpora and emphasize multi-step internal deliberation, yet they lack the ability to interact with external environments or invoke tools. In contrast, agentic models including GLM-4.5 (Zeng et al., 2025), Kimi K2 (Team et al., 2025a), DeepSeek-V3.1 (DeepSeek-AI, 2025), and agent frameworks such as OAgents (Zhu et al., 2025) and BrowseMaster (Pang et al., 2025) prioritize tool use, planning, browsing, and code execution. These systems achieve strong performance in tool-intensive environments, but typically lag behind reasoning-centric models when multi-step logical derivation is required. The divergence in training objectives produces two complementary but disjoint capabilities: internal chain-of-thought reasoning and external environment-driven tool use.

More recently, several hybrid agentic systems attempt to combine reasoning and tools within a single model family. Representative examples include GPT-5 and partially open systems such as Qwen3 (Yang et al., 2025a), which support both reasoning and tool capabilities. However, existing hybrid systems typically treat these behaviors as loosely coupled components. For instance, Qwen3 trains reasoning and agentic abilities in separate alignment stages and lacks a unified controller to decide between reasoning and tool use at inference time. Closed systems such as GPT-5 (OpenAI, 2025b) provide hybrid capabilities but do not disclose data construction, trajectory formats, or training pipelines, making it difficult to assess or reproduce their underlying mechanisms. As a result, current hybrid solutions fall short of achieving integrated decision making that jointly governs instant responses, internal reasoning, and agentic actions.

A parallel line of work aims to regulate the computational overhead of reasoning. Recent approaches explore when models should produce long chains of thought and when they should respond directly, thereby reducing unnecessary computation on easy inputs (Chen et al., 2024b; Fatemi et al., 2025; Lou et al., 2025; Kumar et al., 2025; Sui et al., 2025). These think and non-think methods focus primarily on optimizing the length or triggering of reasoning traces. However, they fundamentally operate within text-only settings and do not address the broader challenge of choosing between internal reasoning and external actions, especially when tool use introduces additional latency and a higher cost of pass (Erol et al., 2025).

To address the challenges mentioned above, we introduce the Adaptive Agent Foundation Model (A²FM), which unifies three execution modes within a single backbone: an *instant* mode for direct answers on simple queries, a *reasoning* mode for long-form chain-of-thought, and an *agentic* mode for multi-step tool use and environment interaction. To bridge the capability gap, A²FM internalizes a self-adaptive router that learns *what to do* per query, naturally combining the strengths of reasoning and agentic paradigms in one model. To address efficiency, we add the *instant* mode so that simple queries are answered directly, avoiding unnecessary reasoning or tool calls. Supervised fine-tuning follows a *route-then-align* principle: the model first performs task-aware routing, then aligns mode-conditioned trajectories under a shared policy.

Nevertheless, simply mixing modes falls short: the model must not only maintain high accuracy but also minimize computational cost, yet boundary queries remain difficult to route and data often underutilized. Prior work either omits explicit modeling of mode mixing (Jiang et al., 2025; Zhang et al., 2025), or focuses solely on efficiency by shortening reasoning traces in post-training (Team et al., 2025b; Arora & Zanette, 2025), or fails to jointly optimize routing and trajectory generation in an end-to-end framework (Wu et al., 2025a). We therefore introduce **Adaptive Policy Optimization (APO)**, a reinforcement learning procedure tailored for mode selection. APO builds on group-relative policy optimization (Shao et al., 2024), enforces adaptive sampling across modes through a combination of forced and adaptive rollouts, and introduces a per-query cost-regularized reward with LLM-as-Judge signals (Zheng et al., 2023). The reward explicitly conditions on whether the instant mode can already solve a query, thereby encouraging minimal-effort instant responses on easy inputs while escalating to reasoning or agentic modes only when external evidence or extended deliberation is needed.

Empirically, $A^2FM$ achieves state-of-the-art results across benchmarks: on agentic tasks it obtains 13.4% on BrowseComp, on reasoning it reaches 70.4% on AIME25, and on general tasks it delivers 16.7% on HLE. Beyond raw accuracy, APO substantially improves efficiency: on SuperGPQA, the adaptive execution yields a cost of pass of only **$0.00487** per correct answer—cutting cost by **45.2%** relative to reasoning and **33.5%** relative to agentic—while maintaining comparable accuracy.

In summary, our key contributions include:

- We present $A^2FM$, the first hybrid reasoning and agentic model that jointly integrates task-aware routing and mode-specific trajectories under a shared backbone.
- We propose A two-stage process: (i) supervised *route-then-align* fine-tuning for mode-conditioned trajectories; (ii) *APO* for mode selection with adaptive sampling and cost-regularized rewards.
- We empirically demonstrate that $A^2FM$ achieves state-of-the-art results at the 32B scale across agentic, reasoning, and general benchmarks, with substantial reductions in token usage and computation versus mode-forcing baselines.

## 2 RELATED WORK

A growing body of work shows that indiscriminate chain-of-thought (CoT) reasoning imposes substantial computational overhead, latency, and token usage on easy inputs (Chen et al., 2025b; Fatemi et al., 2025). Though "auto-thinking" has emerged as a remedy, practical deployments often still rely on manual controls—e.g., Qwen3 requires explicit thinking mode toggling by users (Yang et al., 2025a). To address these issues, recent progress in efficient LLM responses has converged on two core directions, complemented by "when-to-think" research.

### 2.1 LENGTH-AWARE APPROACHES

Length-aware approaches aim to reduce redundant token consumption by training models to prioritize concise CoT when full-step reasoning is unnecessary. This direction adopts two dominant technical paths: (1) Reinforcement Learning with length regularization, which integrates length-regularized rewards to modulate CoT depth and balance accuracy and brevity (Arora & Zanette, 2025; Yeo et al., 2025; Aggarwal & Welleck, 2025); (2) Supervised Fine-Tuning with CoT compression, which builds variable-length CoT via post-reasoning CoT compression or the acquisition of compressed CoT data in the course of reasoning (Xia et al., 2025; Kang et al., 2024; Munkhbat et al., 2025; Liu et al., 2024). By learning to budget inference, these methods reduce token expenditure.

### 2.2 CAPABILITY-AWARE ROUTING APPROACHES

Capability-aware routing enables adaptive mode selection by estimating a model's knowledge boundary via internal signals (uncertainty, logit margins, intermediate representations) to trigger adaptive reasoning modes (Chen et al., 2024a). Self-routing leverages these signals to assess task difficulty and choose between short and long CoT responses (He et al., 2025). Yet, current routers

often use linear probes on internal features and binary policies, limiting sensitivity to nonlinear state–difficulty ties and domain-specific control (e.g., math, code).

A complementary sub-direction is "when-to-think" policies, which explicitly learn whether to trigger reasoning. Several methods introduce new RL reward designs to regulate thinking preferences (Lou et al., 2025; Yu et al., 2025). Despite promising results, they have two key limitations: (i) reliance on complex reward functions or manual annotated data—introducing subjectivity and hindering scalability; (ii) hyperparameter sensitivity and training instability, where misconfiguration causes mode collapse and undermines adaptive reasoning. To mitigate these, (Yang et al., 2025b) propose Bimodal Policy Optimization (BPO), which contrasts thinking vs. non-thinking trajectory utilities for the same input to learn robust switching. In a related vein, (Jiang et al., 2025) propose the Large Hybrid Reasoning Model (LHRM)—it dynamically decides whether to invoke extended reasoning via query semantics and context, but lacks agentic tool-use integration.

# 3 METHOD

## 3.1 PROBLEM FORMULATION

We consider a system that processes queries $x \in \mathcal{X}$ drawn from a task mixture $\mathcal{D}$ and produces outputs $y \in \mathcal{Y}$. For each query, a *router* selects one of three execution modes:

$$\mathcal{M} = \{\texttt{instant}, \texttt{reasoning}, \texttt{agentic}\}, \quad m \sim \pi_{\text{route}}(m \mid x)$$

where $\pi_{\text{route}}$ is a routing policy over the mode set $\mathcal{M}$.

Given the selected mode $m$, the system then generates an output through a *mode policy* $\pi_m$, i.e.,

$$y \sim \pi_m(y \mid x)$$

Each mode induces a distinct form of trajectory $\tau_m$: direct answers for $\texttt{instant}$, chain-of-thought sequences for $\texttt{reasoning}$, and tool-interaction traces for $\texttt{agentic}$. We denote the decoding function of a trajectory under mode $m$ as $f_m(x, \tau_m)$.

Let $\text{Acc}(x, y) \in [0, 1]$ be a task-specific accuracy metric. The expected performance of mode $m$ on input $x$ is

$$Q_m(x) = \mathbb{E}_{\tau_m \sim \pi_m(\cdot \mid x)}\Big[\text{Acc}\big(x, f_m(x, \tau_m)\big)\Big]$$

Our objective is to jointly optimize the router and mode policies such that the system maximizes expected accuracy across the task mixture:

$$\max_{\pi_{\text{route}}, \{\pi_m, f_m\}} \mathbb{E}_{x \sim \mathcal{D}}\left[ \sum_{m \in \mathcal{M}} \pi_{\text{route}}(m \mid x) \, Q_m(x) \right]$$

This unified formulation makes explicit the two levels of decision-making: (1) the router $\pi_{\text{route}}$ decides which mode to activate for a given query, and (2) the mode-specific policy $\pi_m$ determines the quality of the generated trajectory. Together, they provide the foundation for both supervised alignment and reinforcement learning.

## 3.2 STAGE 1: ROUTE-THEN-ALIGN FINE-TUNING

**Training Data Generation.** The first stage centers on supervised route-then-align training, where the model learns to classify a query into one of three modes—$\texttt{instant}$, $\texttt{reasoning}$, or $\texttt{agentic}$—and then generate mode-consistent trajectories. Mode routing is triggered by paired `<classification>` tags, after which the model proceeds with different behaviors:

- In the $\texttt{instant}$ mode, the model directly outputs the final prediction within `<answer>` tags, minimizing deliberation.

- In the $\texttt{reasoning}$ mode, the model provides a chain-of-thought wrapped in `<reasoning>` tags, followed by the final result in `<answer>` tags.

- In the $\texttt{agentic}$ mode, the model interleaves high-level reasoning with external tool usage. While inspired by Agent Foundation Models (Li et al., 2025b), our design departs significantly in how **Plan** and **Summary** are used.

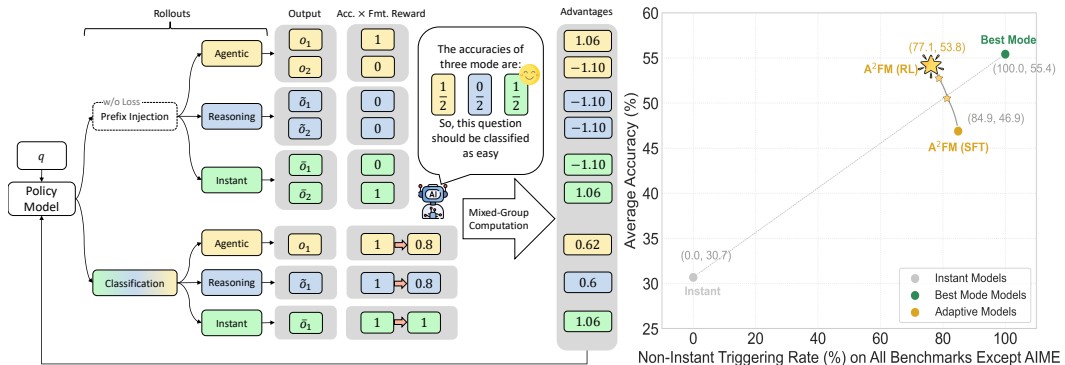

Figure 3: Overview of Adaptive Policy Optimization (APO). **Left:** Rollout and reward process. For each query, mode-specific rollouts are generated either by prefix injection (forced agentic/reasoning/instant) or by adaptive classification. Both prefix-injection tokens and tool-response tokens are excluded from loss since they are not model-generated. **Right:** Accuracy–efficiency trajectory under APO, showing how $A^2$FM progressively approaches the Pareto frontier by improving accuracy while reducing non-instant triggering (excluding AIME24/25).

- **Plan:** appears only once at the beginning, decomposing the query into multiple sub-goals that can be executed in parallel.

- **Summary:** operates dynamically across the process, allowing the system to concurrently aggregate solved sub-tasks, terminate completed threads, and open new ones when needed.

This explicitly parallel architecture enables simultaneous multi-tool execution, substantially improving both efficiency and effectiveness of tool usage.

External tools follow the MCP format and include (see Appendix A.1):

- **Search Tool:** a search engine for information retrieval.

- **Crawl Tool:** a visitor to access web pages.

- **Code Interpreter Tool:** a sandboxed environment to execute Python code.

The `agentic` trajectory begins with a `<plan>` section, executes $N$ tools in parallel (each wrapped in `<tool_call>` tags), and collects their results within `<tool_response>` tags. These tool results are masked during training—following the strategy of Search-R1 (Jin et al., 2025)—so that during the fine-tuning stage, the model focuses on reasoning and routing, not memorization of tool outputs. A final `<summary>` may be added before producing the `<answer>`. Details of the training data trajectories template are deferred to Appendix A.2.

**Data Curation.** To construct diverse, challenging training data, we use two heuristics (difficulty-based sampling adjustment and classification-ambiguous query handling); detailed implementation and rationale are deferred to Appendix B.1.1.

For distillation, we adopt mode-specialized teachers: DeepSeek R1 (Guo et al., 2025) (strong reasoning) for the reasoning mode, and DeepSeek V3.1 (DeepSeek-AI, 2025) (broad general competence) for agentic/instant mode trajectories. This complementary design ensures each mode leverages the most suitable teacher, enabling stronger distillation and more reliable alignment under a shared backbone.

### 3.3 STAGE2: ADAPTIVE POLICY OPTIMIZATION

Aiming to tackle the challenges in balancing accuracy and efficiency in query processing, we propose Adaptive Policy Optimization (APO), which equips the model with dynamic mode-selection—choosing among agentic, reasoning, or instant—based on task demands.

APO builds on GRPO (Shao et al., 2024) but extends it in two key ways: (i) a tailored rollout strategy that ensures sufficient exploration of all modes, avoiding under-sampling; and (ii) an adaptive reward that explicitly encodes the accuracy–efficiency trade-off, rather than relying on coarse binary rewards. This design enables the model to not only maintain correctness but also minimize unnecessary computation, achieving accurate yet cost-effective routing.

**Rollout Strategy.** For each query, APO performs both *mode-forced rollouts* and *adaptive rollouts*. In the forced setting, the model is compelled to operate in each of the three modes—agentic, reasoning, and instant—for $\rho$ rollouts per mode, where $\rho$ is a tunable hyperparameter. Mode enforcement is implemented via prefix injection, in which a pre-specified classification tag is inserted at the beginning of the model's response (see Appendix B.3). This guarantees that every query is explored under all modes, enabling an unbiased estimate of their relative success rates. Such statistics provide the foundation for adaptive rewards (See Adaptive Reward below), which encourages the use of more accurate modes while preferentially selecting the instant mode whenever possible to minimize token consumption.

In addition, APO samples $\gamma$ *adaptive rollouts*, where $\gamma$ is another hyperparameter controlling the number of trials in which the model autonomously selects its operating mode. These adaptive samples provide the opportunity to reward correct self-routing, thereby reinforcing accurate and cost-efficient mode selection.

**Accuracy Reward.** We adopt the LLM-as-Judge framework (Zheng et al., 2023) (See the judge prompt in Appendix I.3) to assess correctness, where a judge model $M_j$ provides binary feedback:

$$r_{\text{acc}} = \mathbb{I}[M_j(x, \hat{y}) = 1].$$

This avoids the limitations of rule-based metrics (e.g., F1, EM) that cannot fully capture the validity of open-ended outputs.

**Adaptive Reward.** To further guide proper mode selection, we introduce an adaptive reward that explicitly favors minimal-effort solutions on easy tasks. Intuitively, if a query can already be solved by the instant mode, then choosing a more costly mode (reasoning or agentic) should incur a penalty. Formally, we label a query as easy if the instant mode achieves accuracy above a threshold $\tau$. For such easy queries, the reward is defined as

$$r_{\text{adaptive}} = \begin{cases} 1 - p^{\alpha}, & \text{if a non-instant mode is chosen}, \\ 1, & \text{otherwise}, \end{cases}$$

where $p$ is the empirical success rate of all forced rollouts on that query, and $\alpha > 0$ is a scaling factor. This design ensures that correct use of the instant mode always receives full reward, while reasoning or tool use on easy tasks is penalized in proportion to how confidently the query could be solved instantly. For hard queries, no penalty is applied, prioritizing the focus on correctness.

**Format Reward.** We enforce strict schema compliance: if the output violates the mode-specific format (e.g., tool tags in instant), the reward is zero; otherwise it is one:

$$r_{\text{format}} = \begin{cases} 1, & \text{if } y \text{ matches the format of mode } m, \\ 0, & \text{otherwise}. \end{cases}$$

**Total Reward.** The final reward combines accuracy, adaptivity, and format constraints:

$$r_{\text{total}} = r_{\text{accuracy}} \times r_{\text{adaptive}} \times r_{\text{format}}.$$

This multiplicative design ensures that failure in any single component (e.g., wrong answer, misuse of modes, or format violation) immediately results in reward deduction, thereby enforcing strict correctness while still encouraging efficiency and proper schema adherence.

We adopt GRPO's objective calculation with certain specializations. Specifically, we strictly employ on-policy training, as varying model capabilities may affect query classification, thereby leading to an unstable training process. Additionally, to expedite training and explore more efficient mode selections, we omit the computation of KL divergence. Consequently, given an actor parameterized by $\theta$, for each question $x_i$, APO samples a group of outputs $\{y_{i1}, y_{i2}, \cdots, y_{iG}\}$ from the old policy

$\pi_{\theta_{\text{old}}}$ and then optimizes the policy model by maximizing the following objective:

$$J_{\text{APO}}(\theta) = \mathbb{E}_{[xi \sim P(Q), \{y_{ij}\}_{j=1}^{G} \sim \pi_{\theta_{\text{old}}}(Y|xi)]}$$

$$\frac{1}{G}\sum_{t=0}^{G}\frac{1}{|y_{ij}|-1}\sum_{t=0}^{|y_{ij}|-1}\min\left\{\frac{\pi_\theta\left(o_t^{(ij)} \mid s_t^{(ij)}\right)}{\pi_{\theta_{old}}\left(o_t^{(ij)} \mid s_t^{(ij)}\right)}\hat{A}_{ij}, \text{clip}\left(\frac{\pi_\theta\left(o_t^{(ij)} \mid s_t^{(ij)}\right)}{\pi_{\theta_{old}}\left(o_t^{(ij)} \mid s_t^{(ij)}\right)}, 1-\varepsilon, 1+\varepsilon\right)\hat{A}_{ij}\right\}$$

where $o_t^{(ij)}$ is the token at index $t$ in response $y_{ij}$; $s_t^{(ij)} := (x_i, a_0^{(ij)}, ..., a_{t-1}^{(ij)})$ is the prefix context when generating $a_t^{(ij)}$; $G := 3 \times \rho + \gamma$ is the sum of all mode trajectory in the rollout stage.

This improved framework with rollout strategy and adaptive reward ensures more stable policy optimization and targeted mode selection. Through this refined reinforcement learning paradigm, A$^2$FM achieves enhanced performance in adaptive mode routing, while simultaneously boosting overall accuracy and efficiency across diverse query scenarios.

# 4 EXPERIMENT

## 4.1 EXPERIMENTAL SETUP

We train A$^2$FM on a mixture of agentic, reasoning, and code-related datasets, while also retaining relatively simple cases to enable learning of the instant mode. Full dataset composition and sampling strategy are provided in Appendix B.1.3. For comparison, we benchmark against three categories of systems: (i) general-purpose LLMs, (ii) agent frameworks, and (iii) agent foundation models at the 32B scale. A complete list of baselines is detailed in Appendix B.2.

**Evaluation Benchmarks.** We evaluate our model on three key types of benchmarks. First, for agentic tasks, we use BrowseComp(Wei et al., 2025a), GAIA(Mialon et al., 2023) (we test just a 103 text-only subset extracted in (Wu et al., 2025b)), and XBench-DeepSearch(Chen et al., 2025a), which assess the model's ability to interact with external tools like search engines and databases for information seeking. Second, for reasoning-dependent tasks, we utilize AIME24(of America , MAA), AIME25(of America , MAA), and Math500(Hendrycks et al., 2021), which focus on mathematical reasoning and problem-solving without relying on external information. Finally, for general knowledge and comprehension, we use GPQA-d(Rein et al., 2024), SuperGPQA(Du et al., 2025), MMLU-Pro(Wang et al., 2024), and HLE(Phan et al., 2025) (We use a 500 text-only subset defined in (Li et al., 2025c)), which test the model's ability to synthesize information and apply broad general knowledge across a wide range of domains.

**Implementation Details.** We conduct all experiments on the Qwen2.5-32B-Instruct backbone. In the route-then-align SFT stage, we train for 3 epochs with a batch size of 256, AdamW optimizer, cosine decay learning rate schedule, and a max sequence length of 32,768. In the APO stage, we train for 2 epochs with a learning rate of 1e-6, 5 warmup steps, batch size 128, and 12 rollouts per prompt (with $\rho = 3$ and $\gamma = 3$) capped at 65,536 tokens. We set $\alpha = 2$ for adaptive reward calculation. For inference, we set temperature 1.0, top-p 0.9, top-k 20, and a max output length of 131,072.

## 4.2 MAIN RESULTS

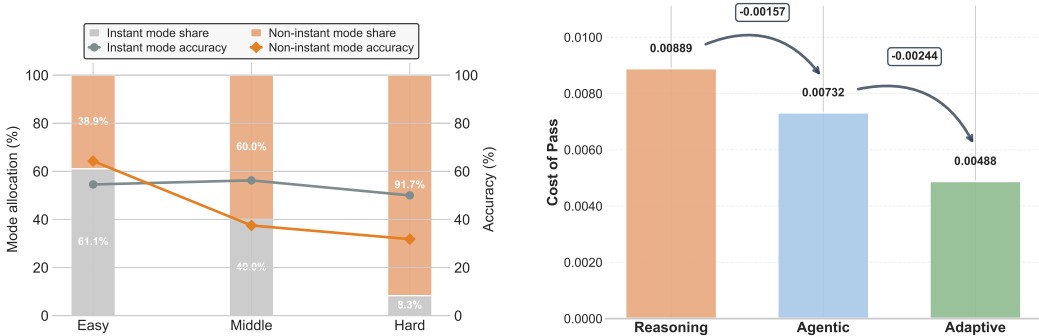

(a) Relation between task difficulty, allocation ratio, and accuracy for instant and non-instant modes.

(b) Cost of Pass from different modes.

Figure 4: Efficiency analysis on SuperGPQA: mode allocation vs. difficulty and token usage across modes.

**Evaluation of Agentic Abilities.** The results in Table 1a show that our adaptive mode (A$^2$FM) achieves consistently competitive performance across all agentic benchmarks. On XBench-DS, A$^2$FM reaches **56.0%**, the

(a) Agentic benchmarks (GAIA, BrowseComp, XBench-DeepSearch).

| Benchmark | A$^2$FM | A$^2$FM Agentic | GPT-4.1 | Claude 4 Sonnet | Qwen2.5 Instruct-32B | OAgents (GPT-4.1) | DeepDive | WebSailor | Asearcher | AFM Search |
|---|---|---|---|---|---|---|---|---|---|---|
| XBench-DS | **56.0** | 54.0$^{-2.0↓}$ | 17.0 | 21.5 | 8.7 | 47.0 | 50.5 | 53.3 | 42.1 | 54.0* |
| GAIA | 57.3 | **60.7**$^{+3.4↑}$ | 22.3 | 22.3* | 13.6 | 58.3 | - | 53.2 | 52.8 | 55.3 |
| BC | 13.4 | 14.4$^{+1.0↑}$ | 1.5 | 2.6 | 0.6 | 13.7 | **14.8** | 10.5 | 5.2 | 11.1 |

(b) Reasoning benchmarks (MATH500, AIME24, AIME25).

| Benchmark | A$^2$FM | A$^2$FM Reasoning | GPT-4.1 | Claude 4 Sonnet | o1 | Simple TIR | Effective TIR | Auto TIR | ReTool | AFM Code |
|---|---|---|---|---|---|---|---|---|---|---|
| **MATH500** | 95.0 | 95.2$^{+0.2↑}$ | 92.4 | 94.0 | **96.4** | 92.9 | 86.4 | 62.6 | 93.2 | 94.6 |
| **AIME24** | **74.5** | **74.5**$^{+0.0}$ | 46.5 | 43.4 | 74.3 | 59.9 | 42.3 | 33.3 | 67.0 | 66.7 |
| **AIME25** | 70.4 | 70.4$^{+0.0}$ | 37.0 | 33.1 | **79.2** | 49.2 | 25.2 | 16.7 | 49.3 | 59.8 |

(c) General-knowledge benchmarks (GPQA-d, SuperGPQA, MMLU-Pro, HLE).

| Benchmark | A$^2$FM | A$^2$FM Agentic | A$^2$FM Reasoning | GPT-4.1 | Claude 4 Sonnet | Qwen2.5 Instruct-32B | Qwen3 -32B | QwQ |
|---|---|---|---|---|---|---|---|---|
| **GPQA-d** | 63.1 | 67.7$^{+4.6↑}$ | 64.7$^{+1.6↑}$ | 66.3 | **68.3** | 49.5 | 54.6 | 65.6 |
| **SuperGPQA** | 54.7 | **56.0**$^{+1.3↑}$ | 51.2$^{-3.5↓}$ | 50.8 | 55.7 | 38.8 | 43.2 | 43.6 |
| **MMLU-Pro** | 73.8 | 75.8$^{+2.0↑}$ | 77.0$^{+3.2↑}$ | 81.8 | **83.5** | 69.0 | 72.7 | 76.4 |
| **HLE** | 16.7 | 20.6$^{+3.9↑}$ | 13.4$^{-3.3↓}$ | 3.7 | 5.8 | 3.8 | 8.3 | 8.2 |

Table 1: Unified results across (a) agentic, (b) reasoning, and (c) general-knowledge benchmarks. **Bold** = best; underline = second-best. Teal/Red superscripts indicate gain/loss of the forced mode relative to adaptive A$^2$FM. * indicates results reproduced by us. All numbers are reported as avg@1, except AIME24/25 which use avg@32.

highest among all baselines and surpassing the second-best model, AFM-Search (54.0%), by +2.0 points. On GAIA, A$^2$FM ranks second overall, trailing only OAgents—which leverages GPT-4.1 as its backbone—while still outperforming all general-purpose LLMs (e.g., Claude-4-Sonnet at 2.6%) and all 32B-scale agent foundation models. On BrowseComp, A$^2$FM achieves **12.4%**, ranking third behind DeepDive (14.8%) and OAgents (13.7%), but still exceeding AFM-Search (11.1%) and all general-purpose LLMs.

When forcing the model into the agentic mode (A$^2$FM-Agentic), performance further improves: on BrowseComp it rises to **14.4%**, surpassing OAgents and achieving the second-highest score overall, just behind DeepDive (14.8%); on GAIA it reaches **60.7%**, establishing a new SOTA. This demonstrates that A$^2$FM not only delivers robust adaptive performance but also, when focused on tool-intensive reasoning, can outperform specialized frameworks in complex deep-search scenarios.

**Evaluation of Reasoning.** As shown in Table 1b, A$^2$FM achieves reasoning performance comparable to o1 on MATH500 (**95.0%**), while significantly outperforming all 32B-scale agent foundation models (e.g., ReTool 67.0% on AIME24, AFM Code 59.8% on AIME25). On AIME24, it sets a new SOTA with **74.5%**, and on AIME25 it attains **70.4%**, second only to o1 (79.2%). Compared to Claude 4 Sonnet, A$^2$FM is stronger by +33.3 and +40.2 points on AIME24/25, underscoring reasoning capabilities far beyond even top general-purpose LLMs. Notably, adaptive and reasoning-specific variants yield identical results on AIME24/25, showing that A$^2$FM reliably routes nearly all AIME queries into the reasoning mode.

**Evaluation of General Abilities.** The results in Table 1c show that A$^2$FM delivers highly competitive general-domain performance at the 32B scale. The adaptive mode consistently outperforms instruction-tuned baselines (e.g., Qwen2.5-32B) and even surpasses the reasoning-focused QwQ. Despite starting from a 32B instruct backbone, A$^2$FM achieves large gains: on GPQA-d it improves by +13.6 points over Qwen2.5-32B (**63.1** vs. 49.5), and on SuperGPQA it is +15.9 points higher (**54.7** vs. 38.8), notably surpassing GPT-4.1 (50.8) and Claude 4 Sonnet (55.7).

HLE further highlights the model's strength on integrated tasks requiring both reasoning and tool use. Here, A$^2$FM-Agentic achieves a remarkable **20.6%**, exceeding the next-best baseline (QwQ, 8.2%) by +12.4 points, while the adaptive mode (**16.7%**) still leads all comparable 32B models by a wide margin. The gap between adaptive and agentic variants reflects that some queries are routed into the instant mode, illustrating A$^2$FM's ability to balance effectiveness with efficiency even under challenging integrated settings.

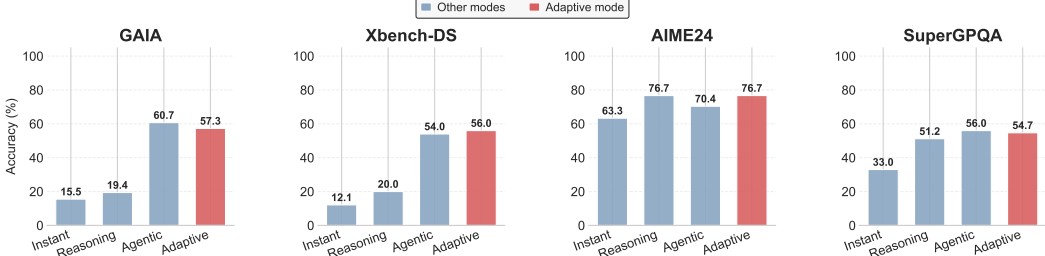

Figure 5: Comparison of adaptive mode (red) against forced single modes across four benchmarks.

**Evaluation of Efficiency.** We focus on $A^2$FM's efficiency by analyzing its mode allocation across task difficulty (Fig. 4a). On SuperGPQA (with human-annotated difficulty), the model uses instant mode for **61.1%** of easy questions, but this drops to just **8.3%** for difficult ones. This shows adaptive routing: it leans on reasoning/tool use for complex tasks and direct answers for simple ones to save computation. Notably, instant response accuracy stays stable at **55%** across all difficulty levels, highlighting this mode's robustness even for harder queries.

We further analyze efficiency through the metric *cost-of-pass* (dollar cost per correct answer) on the four general benchmarks (GPQA-d, SuperGPQA, MMLU-Pro and HLE). Formally, for a model $m$ on problem $p$,

$$\text{Cost-of-Pass}(m, p) = \frac{C_m(p)}{R_m(p)},$$

where $R_m(p)$ is the accuracy and $C_m(p)$ is the expected inference cost, computed from input/output token counts and unit token prices. We follow the official Qwen2.5-32B-Instruct pricing, with \$0.00028 per 1k input tokens and \$0.00084 per 1k output tokens.

As illustrated in Fig. 4b, The reasoning mode incurs a cost of \$0.00889 per correct answer, while the agentic mode reduces this to \$0.00732. Strikingly, the adaptive mode further lowers the cost to only **\$0.00487**, corresponding to a reduction of **45.2%** relative to reasoning and **33.5%** relative to agentic. In other words, adaptive execution delivers each correct answer at roughly **half the cost** of reasoning-based execution.

## 5 ANALYSIS

### 5.1 EFFECTIVENESS OF ADAPTIVE ROUTING MECHANISM

**Accuracy of Routing between $A^2$FM and Human Annotations.** We evaluate routing accuracy on three representative datasets: GAIA (mixed agentic and reasoning), BrowseComp (agentic-dominated), and AIME24 (reasoning-dominated). For GAIA, ground-truth labels are obtained via majority voting from five strong baselines: DeepSeek-R1, DeepSeek-V3.1, GPT-4.1, GPT-5, and Claude-4-Sonnet, and are followed by manual verification. BrowseComp only consists of search-centric queries and is therefore annotated as agentic, while AIME24 contains pure mathematical reasoning problems and is correspondingly annotated as reasoning.

Table 2: Human-labeled mode distribution and model routing accuracy.

| Dataset | Agentic | Reasoning | Instant | Acc. |
|---|---|---|---|---|
| GAIA | 84.5 | 13.6 | 1.9 | 92.2 |
| BC | 100.0 | 0.0 | 0.0 | 94.0 |
| AIME | 0.0 | 100.0 | 0.0 | 100.0 |

As shown in Table 2, our model achieves **92.2%** accuracy on GAIA and **94.0%** on BrowseComp. Errors on GAIA often stem from queries where both reasoning and agentic are plausible (see Case Study in Appendix J.2), while BrowseComp errors arise when the model overestimates its internal knowledge and avoids tool calls. On AIME24, the model reaches **100%**, confirming robust mode discrimination on reasoning tasks.

**Robustness of Adaptive Routing on Complex Tasks.** We compare adaptive routing with forced single-mode execution (agentic, reasoning, and near-instant) across four benchmarks: GAIA, XBench-DeepSearch, AIME24, and SuperGPQA. GAIA and XBench-DeepSearch are primarily agentic-oriented with some reasoning and instant cases; SuperGPQA contain a mix of tasks where tool use and fast responses are critical; AIME24 is reasoning-dominated. Results (Fig. 5) show that adaptive mode achieves competitive or superior performance in most settings. On GAIA, adaptive slightly underperforms pure agentic (57.3 vs. 60.7), primarily due to classification noise. On XBench-DeepSearch and AIME24, adaptive surpasses single modes, demonstrating its strength in routing composite, multi-facet queries to the appropriate execution mode rather than over-committing to a single behavior. On AIME24, adaptive mode strictly matches reasoning mode, since

all queries are routed into reasoning where accuracy dominates. Overall, adaptive mode provides robust performance while retaining flexibility across diverse task types.

**Failure Case Analysis of Adaptive Routing.** Under adaptive routing, $A^2$FM achieves 57.3% accuracy on GAIA, with 44 failed cases in total, among which 5 are routing errors and the remaining 39 arise from within-mode execution (See the confusion matrix in Table 3 ). This decomposition shows that the majority of failures do not stem from the routing mechanism, but from downstream execution within the selected mode.

Table 3: Confusion Matrix for Routing Behavior.

| Pred \ GT | Agentic | Reasoning | Instant |
|---|---|---|---|
| **Agentic** | 84 | 1 | 0 |
| **Reasoning** | 0 | 12 | 0 |
| **Instant** | 3 | 1 | 1 |

Two key observations can be drawn from this matrix: most routing mistakes result from the model over-selecting the instant mode for questions that actually require agentic or reasoning steps, while confusion between agentic and reasoning modes is very rare, a fact that indicates the router already maintains a clear separation between these two modes. All routing failures fall into two patterns: (i) premature instant selection (4 of 5 cases), and (ii) unnecessary agentic invocation for simple factoid or light-reasoning tasks. Illustrative examples are provided in Appendix J.1. It is worth noting that after APO fine-tuning, the model correctly switches to instant for the latter case, thus showing reduced tool-overuse and alignment with the true required complexity.

## 5.2 EFFECTIVENESS OF ADAPTIVE POLICY OPTIMIZATION

**Approaching the Pareto Frontier under APO.** Figure 3 illustrates how $A^2$FM evolves under APO training, evaluated across all benchmarks except AIME24/25, since these tasks are routed entirely into the reasoning mode and offer no meaningful trade-off with instant responses. The three yellow stars mark checkpoints at step 10, 30, and 50, showing a clear trajectory where the adaptive model steadily reduces non-instant usage while improving accuracy. As training progresses, $A^2$FM moves closer to the Pareto frontier of accuracy–efficiency trade-offs, indicating that APO effectively teaches the router to allocate more queries to the instant mode without sacrificing correctness. At convergence, A$^2$FM (RL) achieves an average accuracy of **53.8%** with a non-instant ratio of **77.1%**, compared to the "Best Mode" oracle (accuracy 55.4%, non-instant ratio 100%). Although the accuracy gap is marginal (–1.6 points), our adaptive model lowers non-instant triggering by **22.9** points, demonstrating the substantial improvement in efficiency while maintaining accuracy.

**Ablation of the Adaptive Reward in APO.** To isolate the contribution of the adaptive cost term, it is critical to evaluate APO using a naive accuracy and format reward. As a core component of APO, our adaptive reward enables query-aware routing: it prompts the model to use instant mode for simple queries, and only switch to reasoning or agentic modes when necessary. Without this term, APO collapses into a GRPO-style objective with prefix-enforced multi mode sampling. We have conducted a controlled ablation comparing two variants: (1) APO with accuracy, format and adaptive reward; (2) APO with accuracy and format reward. We tested these two variants

Table 4: Ablation Results of Adaptive Reward in APO on SuperGPQA.

| Variant | Score | Instant Ratio |
|---|---|---|
| w/o adaptive reward | 55.6 | 50.2 |
| **adaptive reward** | 54.7 | 58.6 |

on SuperGPQA, with the experimental results summarized in Table 4. We observe that the adaptive mode on easy queries while still routing to reasoning or agentic behaviors on harder ones. Notably, the overall score decreases only slightly, but the instant-mode usage increases substantially, indicating that the adaptive reward effectively encourages efficiency without sacrificing much accuracy. This experiment directly measures how essential the cost penalty is for inducing efficient routing under identical sampling conditions.

## 6 CONCLUSION

We present A$^2$FM, an Adaptive Agent Foundation Model that unifies agentic, reasoning, and instant modes under a single backbone with a self-adaptive router. By combining route-then-align training with Adaptive Policy Optimization, A$^2$FM achieves both high accuracy and efficiency, matching or surpassing strong baselines across agentic, reasoning, and general benchmarks. Our results highlight the promise of adaptive multi-mode modeling as a scalable path toward efficient and versatile LLM agents.

## 7 ETHICS STATEMENT

This work complies with the ICLR Code of Ethics and has been reviewed by all authors to ensure adherence to ethical research standards.

**Data usage.** All training and evaluation data employed in this study are publicly available from established datasets and benchmarks (e.g., WebDancer, Taskcraft, GAIA, AIME, GPQA, MMLU-Pro). No private, sensitive, or personally identifiable information is included in our datasets, and we performed no data collection involving human subjects.

**Intended use and impact.** A$^2$FM is developed as a research model to advance the study of adaptive routing across agentic, reasoning, and instant modes. The primary contributions are methodological and benchmark-driven; the system is not designed for direct deployment in safety-critical applications. While efficiency gains can reduce computational cost, we caution that misuse of agentic capabilities (e.g., unconstrained tool use) may carry risks if deployed without safeguards. We therefore encourage further research into safe routing strategies and responsible usage.

**Integrity and transparency.** All reported results are based on controlled experiments and reproducible benchmarks. We disclose implementation details and evaluation settings to ensure transparency. All authors affirm that results are reported honestly, without fabrication or selective reporting, and that no financial or non-financial conflicts of interest exist related to this work.

## 8 REPRODUCIBILITY STATEMENT

We have taken deliberate steps to ensure the reproducibility of our findings.

- **Data.** All datasets used for training and evaluation are publicly available benchmarks (e.g., GAIA, BrowseComp, AIME, GPQA, MMLU-Pro). Detailed descriptions of dataset composition, sampling strategies, and mode balancing are provided in B.1.3.
- **Code.** Our full implementation, including the route-then-align SFT pipeline, APO training procedure, and evaluation scripts, will be released upon publication to facilitate reproduction and extension of our results.
- **Experimental Setup.** Hyperparameters, training configurations, rollout strategies, and reward definitions are specified in 4.1. Token pricing and cost-of-pass calculations are explicitly documented to enable consistent efficiency evaluation.
- **Limitations.** While our methods and datasets are fully described, minor deviations may arise from nondeterminism in sampling and stochastic rollouts during APO training.

By providing code, data references, and detailed experimental settings, we aim to make A$^2$FM fully reproducible and to support future research on adaptive agent foundation models.

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

# A    METHOD

## A.1    TOOLS

Our agentic agent utilizes three types of tool to answer the questions: $web\_search$, $crawl\_page$ and $code\_execute$:

- $web\_search$: We interface with the Google Search engine for information retrieval, conducting web searches via the SerpAPI[1] tool. The core SerpAPI configuration parameters are the search query string and the number of results to return. In practice, the tool issues model-generated queries and, by default, retrieves the top 5 results for each query. Each result includes a title, a snippet, and the corresponding Uniform Resource Locator (URL). This setup provides essential support for subsequent analysis and decision-making processes.

- $crawl\_page$: We employed a web-crawling tool authenticated via Jina API[2] and summarized the retrieved content using gpt-5-mini. The tool's core configuration parameters comprise the Uniform Resource Locator (URL), the summary query. Candidate URLs are generated by the model, and the tool uses the Jina API to crawl each URL. Subsequently, we invoke the gpt-5-mini model to produce a summary for each crawled page; the summary prompt is provided in Appendix I.2.

- $code\_execute$: To ensure usability and security, we realize the code sandbox with nsjail[3], a lightweight utility that provisions isolated execution environments for Python code. By leveraging Linux namespace isolation, nsjail hardens file-system boundaries and prevents unauthorized access to host resources. A notable strength of this approach is its compatibility with containerized ecosystems (e.g., Docker), which facilitates seamless migration across diverse training and testing settings. The tool also offers fine-grained resource controls. during the training process, we cap CPU time at 5 seconds and memory at 5 GB to keep code execution strictly bounded.

## A.2    TEMPLATE

As shown in Table 13, we classify the tasks into three modes: Instant mode, Reasoning mode, and Agentic mode. Each mode follows a distinct task execution flow, and we record the task execution trajectories for each mode, which are then used for supervised fine-tuning (SFT) and reinforcement learning (RL) training. The detailed trajectories for each mode are in Table 13.

---

[1] `https://google.serper.dev/search`

[2] `https://jina.ai/`

[3] `https://github.com/google/nsjail`

# B    EXPERIMENTS

## B.1    DATA

### B.1.1    DATA CURATION

To construct diverse and challenging training data, we combine two heuristics. First, a difficulty-based sampling strategy adjusts the natural distribution of task success. In practice, raw sampling tends to yield a U-shaped distribution: tasks are either consistently solved or consistently unsolved, with few in between. To increase coverage of moderately difficult tasks, we deliberately lower the proportion of "always-solved" cases, resulting in a J-shaped distribution (See Fig. 6) that emphasizes more challenging queries. Second, we target classification-ambiguous queries where routing decisions are uncertain. Instead of relying on majority votes, we select the tag corresponding to the mode that achieves the highest accuracy on that query. This strategy emphasizes supervision on decision boundaries, encouraging the model to improve its discrimination ability in cases where multiple modes appear plausible.

### B.1.2    DIFFICULTY DISTRIBUTION

As shown in Fig. 6, our difficulty-based sampling results in a J-shaped distribution. Compared to the raw U-shaped pattern, this distribution reduces the proportion of trivially solved cases and emphasizes more challenging queries, thereby providing better coverage of moderately difficult tasks.

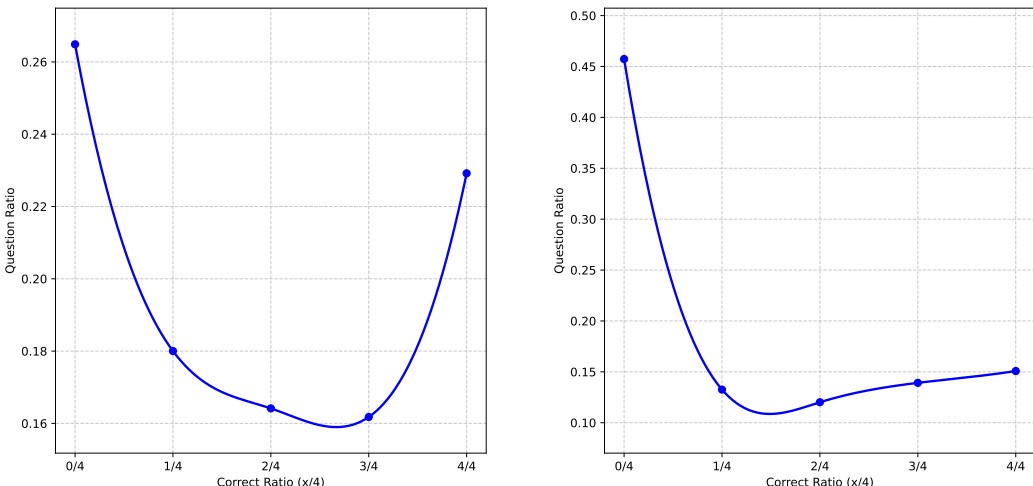

Figure 6: Data distribution among agentic (left) and reasoning (right)

### B.1.3    DATASET COMPOSITION

Our dataset sources encompass a variety of task types and domains. We specifically chose not to focus on easy datasets but rather to select data from more challenging and complex sources. These include datasets such as Webdancer(Wu et al., 2025b), Webshaper(Tao et al., 2025), Taskcraft(Shi et al., 2025), 2Wiki(Ho et al., 2020), MusiqueHo et al. (2020), and HotpotQA(Yang et al., 2018) for agentic-related tasks, and mathematic reasoning datasets like Skywork-Math(Zeng et al., 2024), DAPO-Math-17K(Yu et al., 2025), and GSM8K(Cobbe et al., 2021). Additionally, code-related data sources like Skywork-Code(Zeng et al., 2024) and Codeforces(Penedo et al., 2025) were incorporated for programming tasks. Importantly, while we did not curate separate datasets solely for the instant mode, we deliberately retained relatively simple examples within the search and reasoning datasets. This ensures balance across the three modes and allows the model to naturally learn to apply the instant mode on straightforward tasks. Finally, we created 5289, 2829, and 2890 trajectories for agentic, reasoning and instant modes respectively during the route-then-align SFT phase and 6000 question-answer pairs during the APO phase.

## B.2    BASELINES

We conducted a comprehensive evaluation of our trained model against three categories of systems: (1) general-purpose large language models (LLMs), including closed-source models such as GPT-4.1(OpenAI,

2025a), o1 (Jaech et al., 2024), Claude-4-Sonnet(Anthropic, 2025), and open-source models DeepSeek-R1(Guo et al., 2025), Qwen2.5-Instruct-32B (Yang et al., 2024), Qwen3-32B(Yang et al., 2025a), QwQ-32B(Team, 2025); (2) state-of-the-art agent frameworks, such as OAgents(Zhu et al., 2025); and (3) Agent Foundation Models, including DeepDive(Lu et al., 2025), WebDancer(Wu et al., 2025b), WebSailor(Li et al., 2025a), Asearcher-Web(Gao et al., 2025), AFM-Search(Li et al., 2025b), SimpleTIR(Xue et al., 2025), Effective-TIR(Bai et al., 2025), AutoTIR(Wei et al., 2025b), and ReTool(Feng et al., 2025).

### B.3 ROLLOUT

**Mode-Forcing Strategy** To enforce the model's selection of a specific execution mode for a given task, we utilize a fixed prompt structure that directs the model to choose the appropriate mode. The prompts for each mode are outlined below:

1. **Reasoning Mode:** This task requires complex logical reasoning (such as mathematical proofs, multi-step problem solving) and causal analysis, so I will select `reasoning_mode`. `<classification> reasoning_mode </classification>`

2. **Agentic Mode:** This task requires acquiring real-world information (such as news and data) or executing code (such as programming problems, data processing, or statistics), so I will select `agentic_mode`. `<classification> agentic_mode </classification>`

3. **Instant Mode:** This task needs no real-world info, code, or complex reasoning—just basic knowledge or brief responses, so I will select `instant_mode`. `<classification> instant_mode </classification>`

## C ASSESSING THE RELIABILITY OF THE LLM JUDGE

To assess the reliability of the binary LLM judge used in APO's reward mechanism, we conduct validation across multiple datasets and cross-check the judgments with human evaluation. The main goal is to verify that the chosen judge model (We use GPT-5-Mini in our experiment) produces consistent and unbiased binary decisions under diverse task types, including both deterministic reasoning and open-ended agentic tasks.

We first compare two evaluation paradigms: Exact Matching (EM) and binary LLM-as-Judge. On deterministic benchmarks such as AIME24/25, both methods yield identical correctness labels, showing 100% consistency. However, EM fails on tasks involving free-form or semantically rich outputs (e.g., GAIA, BrowseComp), where equivalent answers may differ lexically. Thus, the binary LLM-as-Judge provides a unified and semantically robust evaluation signal across all datasets.

To further confirm judgment stability, we perform a human verification study comparing GPT-5-Mini and Qwen2.5-72B-Instruct, following the same evaluation setup and standardized prompt defined in Section I.3. Results are summarized in Table 5. GPT-5-Mini exhibits perfect alignment with human assessments across all samples, while Qwen2.5-72B-Instruct shows minor inconsistencies due to formatting ambiguity, confirming GPT-5-Mini's superior reliability for binary reward evaluation.

Table 5: Human evaluation of LLM judge consistency on GAIA.

| Judge Model | Disagreement with Human Judgment |
|---|---|
| GPT-5-Mini | 0/103 |
| Qwen2.5-72B-Instruct | 1/103 |

Overall, these results demonstrate that GPT-5-Mini-as-a-Judge provides a stable and fair binary judgment signal suitable for reinforcement learning supervision, ensuring both semantic robustness and cross-dataset consistency.

## D EFFECT OF TEACHER MODEL CHOICE ON DISTILLED PERFORMANCE

We examine whether the choice of teacher model influences the supervised route-then-align stage of $A^2FM$. Teachers with substantially different raw capabilities are compared across two domains: agentic tasks (GPT-5-Mini vs. DeepSeek-V3.1) and mathematical reasoning (DeepSeek-R1 vs. Qwen3-235B-A22B-Thinking-2507). For each teacher, we measure (i) its raw performance on domain-specific benchmarks and (ii) the performance of an SFT model distilled from that teacher using identical data size, backbone, and training setup.

Table 6 reports the results. In both domains, teachers differ meaningfully in raw accuracy. For example, GPT-5-Mini exceeds DeepSeek-V3.1 by 5.5 points on BC-200 and 7.9 points on GAIA; DeepSeek-R1 exceeds

Qwen3-235B-Thinking by 7.0 points on AIME24. However, their corresponding distilled SFT models converge to nearly the same performance: the agentic SFT scores differ by only 0.9 points, and the reasoning SFT scores differ by only 3.1 points. These findings indicate that the route-then-align framework largely absorbs teacher variability, leading to stable downstream performance even when teacher capability varies significantly.

Table 6: Raw teacher model performance vs. distilled SFT performance.

| Task Domain | Teacher Model | Raw Teacher Score | SFT Score After Distillation |
|---|---|---|---|
| Agentic | GPT-5-Mini | 31.0 (BC-200) / 70.0 (GAIA) | 51.5 |
| | DeepSeek-V3.1 | 25.5 (BC-200) / 62.1 (GAIA) | 52.4 |
| Reasoning | DeepSeek-R1 | 90.2 (AIME24) | 66.7 |
| | Qwen3-235B-A22B-Thinking-2507 | 83.2 (AIME24) | 63.6 |

## E  EVALUATING THE FUNCTIONAL ROLE OF THE INSTANT MODE

We study whether the instant mode is essential for the efficiency gains exhibited by $A^2$FM, and whether comparable behavior could emerge from a two-mode (reasoning + agentic) system. A direct ablation of removing the instant mode within APO is not well-defined. APO's adaptive penalty relies on categorizing queries as "easy" or "hard" based on the success rate of forced-instant rollouts. Without the instant mode, APO lacks the baseline required to infer query difficulty. Thus, a straightforward two-mode APO variant cannot reproduce the adaptive mechanism central to $A^2$FM.

To empirically isolate the contribution of the instant mode at inference time, we design an alternative comparison on a 200-query subset of SuperGPQA. The first setting executes both the reasoning and agentic modes for every query and computes Pass@2 and Cost-of-Pass (CoP) from these two trajectories. This simulates a system without access to the instant mode. The second setting uses the full $A^2$FM router to choose among instant, reasoning, and agentic modes.

Table 7: Comparison of efficiency and performance on a 200-query subset of SuperGPQA.

| Setting | Pass@2 | Cost-of-Pass (CoP) |
|---|---|---|
| Reasoning + Agentic (two modes) | 67.0 | 0.00812 |
| Adaptive routing ($A^2$FM) | 62.5 | 0.00432 |

The results indicate that the joint two-mode system attains slightly higher Pass@2 (+4.5 points), but incurs nearly double the computational cost. Under adaptive routing, the instant mode is selected for more than half of successful Pass@2 cases, and the resulting Cost-of-Pass decreases by 46.8%. These observations show that the instant mode plays a central role in handling simple queries efficiently, reducing unnecessary reasoning or tool-use trajectories while maintaining competitive task performance.

## F  ABLATION ON THE PARALLEL AGENTIC EXECUTION ARCHITECTURE

We analyze the contribution of A²FM's parallel agentic execution design by comparing it against a sequential baseline under matched training conditions. The parallel architecture allows each agentic step to dispatch multiple tool calls concurrently, whereas the sequential baseline follows a standard ReAct-style pipeline in which all calls are serialized (Wu et al., 2025b). Both variants are trained with the same 1,968 trajectories and identical hyperparameters.

The motivation for the parallel design is twofold. First, it establishes a clear capability–cost separation between the agentic mode and the instant mode by assigning the agentic branch a high-capacity, high-cost behavioral profile. Second, it better reflects real-world agentic workloads, where multiple subtasks (e.g., parallel search or crawling) can be processed concurrently, enabling broader evidence collection for complex queries.

We evaluate both variants on GAIA, BrowseComp, and XBench, and measure efficiency using Cost-of-Pass (CoP). Results are shown in Table 8.

Across all benchmarks, the parallel architecture yields consistent performance improvements (+2.9 pp on GAIA, +3.1 pp on BrowseComp, +5.0 pp on XBench), indicating that concurrent tool execution improves evidence gathering and task coverage. As expected, these gains come with moderately higher computational cost. This capability–cost trade-off is essential for APO, which requires modes with distinct efficiency profiles

Table 8: Performance and efficiency of sequential vs. parallel agentic execution.

| Architecture | GAIA | BrowseComp | XBench | Cost-of-Pass (CoP) |
|---|---|---|---|---|
| Sequential baseline | 47.6 | 9.1 | 42.0 | 0.00586 |
| Parallel (ours) | 50.5 | 12.2 | 47.0 | 0.00768 |

to enable meaningful adaptive routing: the parallel agentic mode offers a high-capability option for complex queries, while the sequential and instant modes serve as lower-cost alternatives.

## G  COMPARISON OF $A^2$FM BEFORE AND AFTER APO.

Quantifying the contribution of the RL phase (APO) is essential for understanding $A^2$FM's improvements in performance and efficiency. We thoroughly compare our SFT and APO versions in two aspects: Accuracy and Non-Instant Triggering Rate. Both the model's accuracy at the standalone SFT stage and final APO stage across benchmarks, and its Non-Instant Triggering Rate in these two stages, are detailed in Table 9. We observed that on the one hand, APO yields substantial accuracy gains across all task families: reasoning benchmarks see the largest improvements (e.g., +11 on MATH500, +10 on AIME24, +8.6 on AIME25), while agentic datasets and broad general-knowledge evaluations also show consistent boosts. This proves APO is not a minor refinement to SFT, but the primary source of $A^2$FM's final performance. On the other hand, APO noticeably reduces unnecessary use of expensive non-instant modes: its Non-Instant Triggering Rate drops by over 10 points on average on general benchmarks, and stays unchanged on reasoning-only benchmarks (AIME24/25) as expected. This shows that APO learns cost-aware routing, preferring instant mode when sufficient and invoking heavier modes only when needed, which is exactly as intended by the adaptive reward design.

Table 9: Model Performance at the SFT and APO Stages.

| Benchmark | Accuracy | | NITR | |
|---|---|---|---|---|
| | SFT | APO | SFT | APO |
| XBench | 51.0 | **56.0** | 96.0 | **93.0** |
| GAIA | 51.5 | **57.3** | 96.2 | **93.2** |
| BrowseComp | 10.1 | **13.4** | 95.0 | **89.9** |
| MATH500 | 84.0 | **95.0** | 86.0 | **77.0** |
| AIME24 | 64.5 | **74.5** | 100.0 | 100.0 |
| AIME25 | 61.8 | **70.4** | 100.0 | 100.0 |
| GPQA-d | 56.0 | **63.1** | 88.2 | **77.8** |
| SuperGPQA | 46.9 | **54.7** | 63.8 | **50.2** |
| MMLU-Pro | 65.0 | **73.8** | 58.0 | **46.0** |
| HLE | 11.0 | **16.7** | 96.0 | **89.7** |

## H  GENERALIZATION ACROSS MODEL BACKBONES AND PARAMETER SCALES

We evaluate the portability of the A²FM training framework across multiple backbones (Qwen2.5, Qwen3) and parameter scales (4B, 14B, 32B). Unless stated otherwise, all results reported here are based on SFT-only models to isolate the effect of the training recipe.

### H.1  EXPERIMENTAL SETUP

We consider two families of backbones (Qwen2.5, Qwen3) and three parameter scales (32B, 14B, 4B). At the 32B scale, we directly compare two backbones (Qwen2.5-32B-Instruct and Qwen3-32B). At the 14B and 4B scales, we compare A²FM models with available baselines, including Webthinker (14B), DeepSeekR1 DistillQwen (14B), and Qwen3 base models (reasoning / non-reasoning variants). Evaluation follows the same benchmark suite as in the main experiments: GAIA, BrowseComp, AIME24, AIME25, SuperGPQA, and HLE.

## H.2  32B Backbone Comparison

Table 10 reports the performance of A²FM on two distinct 32B backbones.

Table 10: Performance of A²FM on different 32B backbones (SFT-only).

| Dataset | A²FM-Qwen2.5-32B-Instruct | A²FM-Qwen3-32B |
|---------|---------------------------|----------------|
| GAIA | 51.5 | 55.3 |
| BrowseComp | 10.1 | 12.0 |
| AIME24 | 64.5 | 68.6 |
| AIME25 | 61.8 | 63.5 |
| SuperGPQA | 46.9 | 48.2 |
| HLE | 11.0 | 10.5 |

The Qwen3-32B backbone yields improved performance on 5 of 6 datasets, indicating that the A²FM training procedure generalizes robustly across backbone families.

## H.3  14B Scale Evaluation

At the 14B scale, we compare A²FM-Qwen3-14B with available SFT- or RL-based baselines. Results are shown in Table 11.

Table 11: Performance of A²FM-Qwen3-14B vs. 14B baselines (SFT-only).

| Dataset | A²FM 14B | Qwen3-14B (Non-reasoning) | Qwen3-14B (Reasoning) | DeepSeekR1 DistillQwen-14B | Webthinker 14B |
|---------|----------|---------------------------|-----------------------|----------------------------|----------------|
| GAIA | 46.6 | - | - | - | 33.0 |
| BrowseComp | 9.5 | - | - | - | - |
| AIME25 | 59.8 | 58.0 | 55.7 | 55.7 | - |
| HLE | 8.8 | 4.2 | 4.3 | 4.4 | 7.0 |

A²FM consistently outperforms both Qwen3-14B base models and existing 14B baselines, including substantial gains over Webthinker on GAIA (+13.6 points).

## H.4  4B Scale Evaluation

We further evaluate A²FM-Qwen3-4B on the same benchmark suite; results are shown in Table 12.

Table 12: Performance of A²FM-Qwen3-4B vs. Qwen3-4B base models (SFT-only).

| Dataset | A²FM-Qwen3-4B | Qwen3 4B Instruct | Qwen3 4B Reasoning |
|---------|---------------|-------------------|--------------------|
| GAIA | 38.8 | - | - |
| BrowseComp | 8.9 | 52.3 | 22.3 |
| AIME25 | 57.4 | - | - |
| HLE | 6.6 | 4.7 | 3.7 |

Despite the small parameter scale, A²FM-4B remains competitive and in several cases surpasses larger models and baselines (e.g., outperforming the 14B Webthinker model on GAIA).

# I PROMPTS

## I.1 SYSTEM PROMPT

---

### SYSTEM PROMPT OF A$^2$FM

You are required to solve the task by using one of the three mode options: agentic_mode, reasoning_mode, and instant_mode.

1. **Agent Options**:

(a) agentic_mode: choose this agent if the task needs to **search** and **crawl** real-world / factual information (such as news and data) or **executing code** (such as programming tasks, data processing or statistics).

(b) reasoning_mode: choose this agent if the task requires complex logical **reasoning** (such as mathematical proofs, multi-step problem solving) and causal analysis.

(c) instant_mode: use this agent for simple tasks needing no real-world info, code, or complex reasoning. Instead, just basic knowledge or brief responses.

2. **Trajectory Formulation**:

(a) You should first predict one of the three agents above within the function `<classification> ... </classification>`.

(b) Then you should formulate your thinking and processing trajectory according to the rule of the agent you choose:

    i. **agentic_mode rule**:

    A. Objective:
- Your core goal is to systematically solve user-assigned tasks by:
  - Decomposing the task into clear goals & paths.
  - Executing tools purposefully and efficiently.
  - Advancing all goals in parallel, while keeping each goal's paths sequential.
  - Tracking progress with summaries.
  - Delivering a final confirmed answer only when all goals are resolved.

    B. Execution Requirements:
- Follow a logical order of functions/tools.
- Parallelize independent goals; within each goal, execute paths sequentially as fallbacks.
- Each step must include:
  - thinking (before you execute tools, why this tool/path is chosen).
  - `<tool_call>` execution (with correct parameters).
  - Use results from observations to refine next actions.
  - Ensure no redundant tool calls (don't repeat identical queries).
  - Never assume a goal is completed without explicit verification.
  - Continue advancing all goals until they are resolved.

    C. Functions:
- `<plan>` Function:
  - Role: Decompose the original task into goals and execution paths.
  - Rules:
    * 1–5 parallelizable goals.
    * Each goal has 1–5 paths, executed sequentially as fallback options.
    * Define success criteria for each path.
  - Timing: Only the first step.
  - Format Example:
    ```
    <plan>
    ## Goal 1: [Name]
    - Path 1.1: [Approach]
    - Success: [Criteria]
    - Path 1.2: [Approach]
    - Success: [Criteria]
    ## Goal 2: [Name]
    - Path 2.1: [Approach]
    - Success: [Criteria]
    </plan>
    ```
- `<summary>` Function:
  - Role: Recap execution status and decide next actions.
  - Content:
    * Plan summary (original goals/paths).
    * Execution status for each goal: Completed / In Progress / Blocked.
    * Path analysis (which worked, which failed).
    * Next steps: specify which sub-paths to run in parallel.
  - Timing: Every several steps, occurs when there are enough actions to summarize.
  - Example:
    ```
    <summary>
    ## Plan Summary
    [Brief recap of goals]
    ## Execution Status
    ### Goal 1: [Status]
    - Path Analysis: [...]
    ### Goal 2: [Status]
    - Path Analysis: [...]
    ```

---

```
        ## Next Parallel Sub-Paths
        - Goal 1: Path 1.2
        - Goal 2: Path 2.1
        </summary>
```

- `<tool_call>` Tool:
  - Role: Execute tools to advance goals.
    * web_search: it has only one parameter: query (search statement). Example: `{'id': xxx, 'name': 'web_search', 'arguments': {'query': 'xxx'}}`
    * crawl_page: it has two parameters: url (valid link) and query (info to extract). Example: `{'id': xxx, 'name': 'crawl_page', 'arguments': {'url': 'xxx', 'query': 'xxx'}}`
    * code_execute: it has only one parameter: code (Markdown snippet). Example: `{'id': xxx, 'name': 'code_execute', 'arguments': {'code': 'xxx'}}`
  - Rules:
    * Use **1–10** tools per step (each targeting a distinct task part).
    * Each tool call must have complete, valid parameters.
    * Always prefer verifying accuracy with crawl_page after web_search.
  - Timing: All steps except `<plan>`, `<summary>`, and `<answer>`.
- `<answer>` Function:
  - Role: Deliver the final confirmed answer.
  - Rules:
    * Only after all goals are resolved.
    * Must consolidate results across all goals.
    * Answer language must match task language.
  - Format Example:
    ```
    <answer>
    [Final Answer Content]
    </answer>
    ```

D. Execution Rules (Critical):
  - Parallel Goals, Sequential Paths
  - No Early Termination
  - Result Verification
  - Parallel Functions with Limited workers
  - Final Answer Condition

ii. **reasoning_mode rule**:

A. Trajectory:
  - Reasoning Phase: Output `<reasoning>...</reasoning>` with detailed steps (>1000 words).
  - Answer Phase: Present the final conclusion within `<answer>...</answer>`.

B. Detailed Function Specifications:
  - `<reasoning>` Function
  - `<answer>` Function

C. Notes:
  - Do not return any other functions or tools.
  - Output sequence is always reasoning then answer.
  - Reasoning must exceed 1000 words.

iii. **instant_mode Specification**:

A. Objective:
  - Rapidly solve tasks without tool usage or complex reasoning.
  - Provide clear and relevant answers.

B. Detailed Function Spec:
  - `<answer>` Function only.
  - Executed immediately, no planning or tool-calling.

C. Notes:
  - Must not return other functions or tools.
  - Entire trajectory under 300 words.

3. **Important Tips**:

(a) You should obey the rule of the agent option you choose.

(b) Do not give an answer easily unless you are absolutely sure. The answer should be as concise as possible and avoid detailed descriptions. For example, `<answer>Beijing</answer>`.

## I.2 SUMMARY PROMPT

---

**CASE TRAJECTORY OF POST-RL MODEL ON SAME QUESTION ABOVE.**

Target: Extract all content from a web page that matches a specific web search query, ensuring completeness and relevance. (No response/analysis required.)
web search query: ...
Clues and ideas: ...
Searched Web Page: ...

---

> Important Notes:
> - Summarize all content (text, tables, lists, code blocks) into concise points that directly address the query and clues, and ideas.
> - Preserve and list all relevant links ([text](url)) from the web page.
> - Summarize in three points: web search query-related information, clues and ideas-related information, and relevant links with descriptions.
> - If no relevant information exists, just output "No relevant information."

## I.3 LLM JUDGE PROMPT

> **CASE TRAJECTORY OF POST-RL MODEL ON SAME QUESTION ABOVE.**
>
> You are an evaluation assistant. Please determine if the predicted answer is equivalent to the labeled answer.
> Question: {question}
> Labeled Answer: {labeled_answer}
> Predicted Answer: {pred_answer}
> Are these answers equivalent? Please respond with "Correct" if they are equivalent, or "Incorrect" if they are not equivalent. Do not include any other text.

# J CASE STUDY

## J.1 CASE STUDIES OF ROUTING BEHAVIOR

This section provides representative case studies illustrating the behavior of A$^2$FM's routing mechanism under different query types. Routing errors refer to misalignment between the model-predicted mode (instant/reasoning/agentic) and the ground truth mode required to solve the query. Below, we detail the two most frequent error types, along with illustrative examples and post-optimization improvements. Through quantitative analysis on the GAIA dataset (supported by confusion matrix results) and qualitative inspection of misclassified queries, we identify two primary routing error modes, accounting for over 90% of all routing failures.

### J.1.1 OVER-ASSIGNMENT OF INSTANT MODE

This category captures cases where the model prematurely selects the instant mode despite the task requiring structured reasoning or external information access. These errors typically arise from over-confidence in internal knowledge or failure to account for multi-step operations.

> **ROUTING BEHAVIOR UNDER MODEL TRAJECTORY**
>
> **Query:** I'm making a grocery list for my mom, who is a professor of botany and very strict about categorization. Please extract only the vegetables from the list and alphabetize them as a comma-separated list.
>
> **Ground Truth Mode**: Reasoning.
>
> **Predicted Mode**: This is a straightforward grocery-item classification based on common edible-group definitions (vegetable vs fruit) and doesn't require external data, code execution, or complex reasoning. <classification>instant</classfication>.

### J.1.2 UNNECESSARY INVOCATION OF AGENTIC MODE

The second category includes cases in which the model invokes external tools for queries solvable directly via internal knowledge. These errors inflate computational cost without improving accuracy. APO substantially reduces such cases by penalizing non-instant mode usage on easy queries.

> **CASE TRAJECTORY OF POST-SFT MODEL ON EASY QUESTION.**
>
> *Question*: How many more blocks (also denoted as layers) in BERT base encoder than the encoder from the architecture proposed in Attention is All You Need?
>
> **Classification**
>
> This question asks for a specific architectural comparison (number of blocks/layers in BERT Base vs. the Transformer in 'Attention is All You Need'), which requires precise factual lookup about model definitions. Because it needs accurate, up-to-date model parameter information rather than simple common sense, the agentic_mode — which can query external sources — is the appropriate choice.
> <classification>
> **agentic_mode**
> </classification>

---

**First tool call round**

Now it's step 1, I need to make a plan.
<plan>

Goal 1: Determine the number of encoder blocks (layers) in BERT Base

    Path 1.1: Search authoritative BERT paper and official codebase
- Core approach: Use web_search to locate the BERT paper (Devlin et al., 2018) and the Hugging Face or official BERT model repository, then crawl_page to extract the explicit encoder-layer count from the paper's experimental setup or model configuration file.
- Success: A direct citation or model config line (e.g., "num_hidden_layers: X") explicitly stating the encoder block count for BERT Base is extracted and saved.

    Path 1.2: Query popular NLP model resources and documentation
- Core approach: Use web_search to find Hugging Face model card, PyTorch/Transformer documentation, or TensorFlow model card for BERT Base, then crawl_page to extract the stated number of encoder blocks.
- Success: At least one high-quality model resource (model card or documentation) explicitly lists the encoder block count for BERT Base.

    Path 1.3: Inspect canonical model configuration code (fallback)
- Core approach: Use web_search to locate the canonical configuration file or constructor in a framework (e.g., bert-base-uncased) and crawl_page or code_execute to read the model config and extract the layer count programmatically.
- Success: The layer count is programmatically read from the model config and matches the value from Path 1.1 or 1.2.

Goal 2: Determine the number of encoder blocks (layers) in the Transformer architecture from "Attention is All You Need"

    Path 2.1: Locate and extract from the original paper
- Core approach: Use web_search to find the original paper (Vaswani et al., 2017) and crawl_page to extract the encoder layer count from the experimental setup or model configuration section (e.g., Table/sections describing the base model).
- Success: A direct citation or explicit statement from the paper that gives the encoder block count for the base Transformer model is extracted.

    Path 2.2: Consult authoritative implementations and model cards
- Core approach: Use web_search to find official or widely-used implementations (e.g., TensorFlow official models, Hugging Face "transformer-base" if available, or the original repo) and crawl_page to extract the encoder block count.
- Success: At least one authoritative implementation explicitly lists the encoder block count for the base Transformer model from the paper.

    Path 2.3: Verify via secondary literature or reviews
- Core approach: Use web_search to find reviews, tutorials, or survey papers that summarize the architecture of "Attention is All You Need" and extract the encoder layer count.
- Success: Secondary sources corroborate the encoder block count from the primary paper and provide consistent numbers.

Goal 3: Compute the difference (BERT Base encoder blocks minus Transformer base encoder blocks) and verify accuracy

    Path 3.1: Direct numeric subtraction and cross-check
- Core approach: After Goals 1 and 2 return values, compute difference = (BERT Base encoder blocks) - (Transformer base encoder blocks). Verify consistency across at least two independent sources per model.
- Success: The computed difference is produced and matches across at least two independent sources for each model.

    Path 3.2: Programmatic verification using model configs
- Core approach: Use code_execute to load model configs (or synthetic config files) for BERT Base and the base Transformer and programmatically extract the encoder layer counts, then compute the difference.
- Success: Programmatic extraction yields the same layer counts and the computed difference equals the result from Path 3.1.

    Path 3.3: Cite final difference with provenance
- Core approach: Prepare a concise statement of the difference and list the provenance (paper URLs, model card links) that support the numbers used.
- Success: A short, unambiguous answer stating how many more blocks are in BERT Base than in the base Transformer, with at least two cited sources (one per model).

</plan>
Now it's step 2, I need to make tool_calls. I will advance both goals in parallel. For Goal 1 (BERT Base encoder blocks) the primary path is to locate the BERT paper and its model config; for Goal 2 (Transformer base encoder blocks) the primary path is to locate the 'Attention is All You Need' paper and its model config. I'll perform web searches for each paper/model in parallel to find authoritative sources that state the encoder layer counts. After receiving those search results I'll crawl the specific pages to extract the explicit number of encoder blocks/layers for each model.
<tool_call>
{"name": "web_search", "arguments": {"query": "BERT: Pre-training of Deep Bidirectional Transformers for Language Understanding number of encoder layers"}}
{"name": "web_search", "arguments": {"query": "Attention Is All You Need number of encoder layers base model"}}

---

</tool_call>
<tool_response>

---

**Results for Tool Call 1**

**Query:** "BERT: Pre-training of Deep Bidirectional Transformers for Language Understanding number of encoder layers"
**Total Results:** 5

1. **Source:** arXiv:1810.04805 (J. Devlin et al., 2018, Cited by 143,290)
   *Key Content:* Mentions BERT as a pre-trained deep bidirectional model, with fine-tuning via an additional output layer (encoder layer count not explicitly stated here).

2. **Source:** ACL Anthology: N19-1423 (J. Devlin et al., 2019, Cited by 143,290)
   *Key Content:* Similar to the arXiv version; focuses on fine-tuning performance, no explicit encoder layer count.

3. **Source:** Medium Article
   *Key Content:* Explicitly states: **BERT-Base has 12 transformer layers**, 12 attention heads, and 110 million parameters; BERT-Large has 24 transformer layers.

4. **Source:** Semantic Scholar
   *Key Content:* Summarizes BERT's pre-training of bidirectional representations, no explicit encoder layer count.

5. **Source:** Paperspace Blog
   *Key Content:* Introduces BERT's improvement on fine-tuning techniques, no explicit encoder layer count.

---

**Results for Tool Call 2**

**Query:** "Attention Is All You Need number of encoder layers base model"
**Total Results:** 5

1. **Source:** NeurIPS 2017 Paper (A. Vaswani et al., Cited by 192,399)
   *Key Content:* Explicitly states: **The encoder is composed of a stack of** $N = 6$ **identical layers** (each with multi-head self-attention and feed-forward sublayers).

2. **Source:** arXiv:1706.03762 (Jun 12, 2017)
   *Key Content:* Introduces the Transformer as a replacement for recurrent/convolutional models, no explicit encoder layer count.

3. **Source:** Wikipedia
   *Key Content:* Incorrectly mentions "encoder and decoder were both 8 layers of bidirectional LSTM" (misrepresents Transformer's structure; original paper uses 6 encoder layers).

4. **Source:** Medium Article (3 months ago)
   *Key Content:* Confirms the original paper's architecture: **The encoder stack contains 6 identical layers**, and the decoder stack also contains 6 layers.

5. **Source:** Plain English AI Blog (Jul 12, 2025)
   *Key Content:* States: **The encoder has 6 layers**, each consisting of 2 sublayers (multi-head self-attention + feed-forward network).

---

</tool_response>

**Second tool call round**

I will advance both goals in parallel using the primary path for each: verify the encoder layer counts directly from the original papers (BERT and 'Attention Is All You Need') by crawling the PDFs found in the web_search results. I'll extract the explicit statements for BERT Base's encoder blocks and for the base Transformer's encoder layers, then compute the difference. If the PDFs don't contain the explicit counts, I'll use fallback paths (model cards or official repos) in subsequent steps, but first I'll attempt direct extraction from the paper PDFs. <tool_call>
{"name":"crawl_page", "arguments": {"url": "https://arxiv.org/pdf/1810.04805.pdf", "query": "number of encoder layers BERT Base, number of hidden layers, BERT-base encoder blocks"}}
{"name":"crawl_page", "arguments": {"url": "https://papers.neurips.cc/paper/7181-attention-is-all-you-need.pdf", "query": "Encoder: The encoder is composed of a stack of N = 6 identical layers"}}
</tool_call>
<tool_response>

---

### Results for Tool Call 1

**Page 1** [https://arxiv.org/pdf/1810.04805.pdf] **Summary:**

- **Model architecture:** BERT is a multi-layer bidirectional Transformer encoder. The paper denotes the number of layers (i.e., Transformer blocks) as $L$ and the hidden size as $H$.

- **BERTBASE (commonly "BERT-Base") specifications:**
  - Number of encoder layers / Transformer blocks ($L$): 12
  - Hidden size ($H$): 768
  - Number of self-attention heads ($A$): 12
  - Feed-forward/filter size: $4H = 3072$
  - Total parameters: 110M

- **(For context) BERTLARGE:** $L = 24$, $H = 1024$, $A = 16$, total parameters $\approx$ 340M.

- **Note:** "layers" in the paper explicitly refers to Transformer encoder blocks (i.e., encoder layers).

---

### Results for Tool Call 2

**Page 1** [https://papers.neurips.cc/paper/7181-attention-is-all-you-need.pdf] **Summary:**

- **Exact statement:** "Encoder: The encoder is composed of a stack of $N = 6$ identical layers." (Section 3.1)

- **Encoder layer structure:**
  - Each of the $N = 6$ layers has two sub-layers:
  1. Multi-head self-attention mechanism.
  1. Position-wise fully connected feed-forward network.
  - Residual connections are applied around each sub-layer, followed by layer normalization:
    - Output form: $\text{LayerNorm}(x + \text{Sublayer}(x))$.
  - All sub-layers and embedding layers produce outputs of dimension $d_{\text{model}} = 512$ to facilitate residual connections.

- **Feed-forward sub-layer details (applies to each encoder layer):**
  - FFN formula: $\text{FFN}(x) = \max(0, xW_1 + b_1)W_2 + b_2$
  - Input/output dimensionality: $d_{\text{model}} = 512$
  - Inner layer dimensionality: $d_{\text{ff}} = 2048$
  - Can be seen as two $1 \times 1$ convolutions (position-wise, applied identically at every position).

- **Attention used in encoder:**
  - Multi-head self-attention (encoder self-attention): queries, keys, values all come from the previous encoder layer's outputs; each position may attend to all positions in the previous layer.
  - Multi-head attention configuration used in the paper:
    - $h = 8$ heads
    - $d_k = d_v = d_{\text{model}}/h = 64$

- **Related decoder note (parallel statement):**
  - "Decoder: The decoder is also composed of a stack of $N = 6$ identical layers."
  - Decoder layers include the two sub-layers above plus a third encoder-decoder multi-head attention sub-layer; decoder self-attention is masked to prevent attending to subsequent positions.

- **Embeddings and dimension notes relevant to encoder context:**
  - Input tokens are embedded to vectors of dimension $d_{\text{model}} = 512$.
  - Positional encodings (sinusoidal) of dimension $d_{\text{model}}$ are added to embeddings at the bottoms of encoder/decoder stacks.

- **Model configurations (Table 3) confirming $N = 6$ for both base and big variants:**
  - base: $N = 6$, $d_{\text{model}} = 512$, $d_{\text{ff}} = 2048$, $h = 8$
  - big: $N = 6$, $d_{\text{model}} = 1024$, $d_{\text{ff}} = 4096$, $h = 16$

- *(These points summarize all content in the paper directly related to: "Encoder: The encoder is composed of a stack of $N = 6$ identical layers" and the immediate technical details of those encoder layers.)*

</tool_response>

---

**Answer**

I advanced both goals in parallel: (1) determine the number of encoder blocks (layers) in BERT Base, and (2) determine the number of encoder blocks in the Transformer base model from 'Attention is All You Need'. I used crawl_page to extract authoritative statements from the original papers: the BERT arXiv PDF (Devlin et al., 2018) explicitly states BERT-Base has L = 12 encoder layers, and the 'Attention is All You Need' NeurIPS PDF (Vaswani et al., 2017) states the encoder is composed of a stack of N = 6 identical layers. With both verified, I computed the difference: 12 - 6 = 6. I will now provide the final concise answer with provenance.

> **Answer**
>
> \<answer\>
> BERT Base has 6 more encoder blocks/layers than the base Transformer encoder from 'Attention is All You Need' (BERT-Base: 12 encoder layers; 'Attention is All You Need' base encoder: N = 6 encoder layers). Sources: BERT paper (Devlin et al., 2018; arXiv:1810.04805) and 'Attention is All You Need' (Vaswani et al., 2017; NeurIPS 2017 PDF).
> \</answer\>

**CASE TRAJECTORY OF POST-RL MODEL ON SAME QUESTION ABOVE.**

*Question*: How many more blocks (also denoted as layers) in BERT base encoder than the encoder from the architecture proposed in Attention is All You Need?

**Classification**

This is a straightforward factual/comparison question about known transformer architectures (BERT base vs. original 'Attention is All You Need' encoder). It doesn't require web search or code execution; the information is common knowledge. Therefore the instant_mode, which handles basic factual queries, is appropriate.
\<classification\>
**instant_mode**
\</classification\>

**Answer**

I recall that BERT base has 12 transformer encoder layers, while the original 'Attention is All You Need' paper used 6 encoder layers for their base model. So the difference is 12 - 6 = 6 layers.

> **Answer**
>
> \<answer\>
> 6 more layers
> \</answer\>

## J.2 AMBIGUOUS TASKS WITH MULTIPLE VALID MODES

Certain tasks naturally support more than one valid solution pathway, allowing both internal reasoning and external retrieval. These ambiguous cases demonstrate the flexibility of $A^2$FM: the model may correctly answer the query via either the reasoning mode or the agentic mode, depending on context.

A representative example is the following high-level mathematical question:

> *"What is the maximum Hausdorff dimension of a Sidon set in the reals between 0 and 1?"*

This problem admits two legitimate solution routes. First, it can be resolved entirely through internal mathematical derivation, relying on properties of Sidon sets, sumset structure, and fractal dimension arguments—making the reasoning mode appropriate. Second, the answer is also explicitly documented in external mathematical literature, allowing the agentic mode to retrieve the correct value via search or document access.

Both pathways yield the same correct conclusion, and $A^2$FM is capable of solving the query under either mode. These cases highlight that ambiguity in mode assignment is inherent to the task itself, rather than a failure of the routing mechanism.

## K TEMPLATE

## L USE OF LLMS

In the process of writing this paper, large language models (LLMs) were employed solely as an auxiliary tool for expression polishing, with no involvement in core research parts such as study design, data collection, experimental analysis, or conclusion derivation.

After the LLM-assisted polishing, all content of the paper was strictly reviewed and verified by the authors. The authors confirm that the scientific validity, accuracy of experimental results, and originality of the research conclusions were not affected by the LLM tool, and that the core intellectual contribution of the study remains entirely with the research team. We ensure its application complies with the academic ethics guidelines of the relevant field and the usage agreement of the tool itself.

Table 13: Templates used in training

| Instant mode | Thinking mode | Agentic mode |
|---|---|---|
| <classification> instant </classification> | <classification> reasoning </classification> | <classification> agentic </classification> |
| <answer> {response} </answer> | <reasoning> {reasoning_content} </reasoning> | <plan> {plan_content} </plan> |
| | <answer> {response} </answer> | <tool_call> {tool_dicts} </tool_call> |
| | | <tool_response> {tool_obs} <tool_response> |
| | | <summary> {summary_content} </summary> |
| | | <answer> {response} </answer> |

