# OpenReview forum: "A$^2$FM: An Adaptive Agent Foundation Model for Tool-Aware Hybrid Reasoning"
_ICLR.cc/2026/Conference — ICLR 2026 Poster_

### Official Review · Reviewer_2HMU · 2025-10-17

**Soundness:** 3
**Presentation:** 3
**Contribution:** 2
**Rating:** 6
**Confidence:** 5

**Summary:**

This paper introduces the Adaptive Agent Foundation Model ($A^{2}FM$), a framework designed to unify reasoning-centric and agentic large language models (LLMs). The authors argue that existing models are inefficient, often overusing complex reasoning or external tools for simple queries.

The proposed solution, $A^{2}FM$, integrates three distinct execution modes within a single backbone:

For tasks requiring complex chain-of-thought processes.
For tasks that necessitate the use of external tools like web search or a code interpreter.
For simple queries that can be answered directly, minimizing computational cost.


The paper's primary contributions are:

1. $A^{2}FM$ Framework: The first model to integrate agentic, reasoning, and instant modes under a unified backbone with a self-adaptive router.
2. Adaptive Policy Optimization (APO): A novel reinforcement learning method with adaptive sampling and a cost-regularized reward system to jointly optimize for accuracy and efficiency.
3. Demonstrated Efficiency: The adaptive execution model significantly reduces the cost per correct answer by 45.2% compared to the reasoning-only mode and 33.5% compared to the agentic-only mode, while maintaining comparable accuracy

**Strengths:**

The core strength of $A^{2}FM$ is its integration of three distinct operational modes (reasoning, agentic, instant) under a single, self-adaptive router. While prior work focused on switching between short and long chain-of-thought, this paper is novel in its inclusion of tool-use as a third, distinct modality.

The proposed Adaptive Policy Optimization (APO) is a creative and original extension of reinforcement learning for this specific task. It builds upon existing methods but introduces a tailored rollout strategy and a cost-regularized reward that explicitly teaches the model to balance accuracy with computational efficiency, which is a significant methodological advance.

The paper goes beyond simply reporting scores. It includes a deep analysis of the model's behavior, such as the accuracy of its mode routing on human-labeled data , its performance evolution during APO training , and a clear breakdown of how mode allocation changes with task difficulty.

**Weaknesses:**

The paper presents a two-stage training process (SFT then APO) but only reports the final performance . A crucial missing experiment is an evaluation of the model's performance after only the supervised fine-tuning (SFT) stage, before APO is applied. This would quantify the exact performance and efficiency gains attributable to the reinforcement learning phase, which is a central contribution.

The paper's primary motivation is to improve efficiency by avoiding overthinking on simple queries via an "instant mode". However, this claim is not tested directly. An ablation study training a two-mode variant (reasoning and agentic only) using the same APO framework would be highly informative. Comparing this two-mode model's performance and cost-of-pass against the three-mode $A^{2}FM$ would provide direct evidence for the efficiency gains provided by the instant mode.

The APO methodology introduces an adaptive, cost-regularized reward to balance accuracy and efficiency . The paper would be stronger if it included an experiment where the model is trained with APO using a reward function that only considers accuracy. This would demonstrate how essential the cost penalty is for learning efficient routing behavior.

**Questions:**

The paper presents a complex, multi-stage framework, and its success is impressive. To better isolate the impact of each novel component, could you provide additional details or ablation studies?

The primary contribution is the Adaptive Policy Optimization (APO) stage . To quantify its specific contribution, could you report the performance and efficiency (cost-of-pass) of the model after only the supervised "route-then-align" SFT stage? This would create a clear baseline to measure the exact improvements gained from the reinforcement learning phase.

The efficiency argument hinges on the "instant mode" preventing overthinking on simple queries. To validate this, have you considered an ablation where you train a two-mode variant ($A^{2}FM_{no-instant}$) with only the reasoning and agentic modes? Comparing its cost-of-pass to the full three-mode model on a benchmark with a mix of difficulties (like SuperGPQA) would provide direct evidence of the instant mode's value.

APO uses a cost-regularized reward to balance accuracy and efficiency. How critical is this specific reward design? What happens if the model is trained using APO with a reward function that only considers accuracy ($r_{acc}$)? Does the model learn to default to more powerful (and costly) modes, or does it still achieve some level of efficiency?

The APO stage involves several key hyperparameters, such as the number of forced vs. adaptive rollouts ($\rho=3$, $\gamma=3$) and the adaptive reward scaling factor ($\alpha=2$). Could you comment on the model's sensitivity to these choices? A brief analysis would help future practitioners understand the robustness and tuning requirements of the APO method.

The accuracy reward is determined by an "LLM-as-Judge" . Could you please specify which model was used in this role? Moreover, a brief discussion on the potential biases or failure modes of this judge model would be valuable, as its reliability is critical to the quality of the reward signal during RL training.

---

> ### Author Response · Authors · 2025-11-19
>
> **W1:**
> >The paper presents a two-stage training process (SFT then APO) but only reports the final performance . A crucial missing experiment is an evaluation of the model's performance after only the supervised fine-tuning (SFT) stage, before APO is applied. This would quantify the exact performance and efficiency gains attributable to the reinforcement learning phase, which is a central contribution.
>
> **Q2:**
> >The primary contribution is the Adaptive Policy Optimization (APO) stage . To quantify its specific contribution, could you report the performance and efficiency (cost-of-pass) of the model after only the supervised "route-then-align" SFT stage? This would create a clear baseline to measure the exact improvements gained from the reinforcement learning phase.
>
> **Response:**
>
> Thank you for pointing out this important missing ablation. We agree that quantifying the contribution of the RL phase (APO) is essential for understanding A²FM’s improvements in both **performance** and **efficiency**.
>
> Although Figure 3 in the paper reported the *average* SFT performance (46.9 across benchmarks except AIME24/25), space constraints prevented us from presenting the full per-benchmark comparison. We provide the complete data below and will incorporate a condensed version into the camera-ready.
>
> We will thouroughly compare our SFT and APO versions in two aspects: Accuracy and Non-Instant Triggering Rate.
>
> **Accuracy across benchmarks**
>
> | Benchmark     | SFT      | APO (final) |
> |---------------|----------|-------------|
> | XBench        | 51.0     | **56.0**    |
> | GAIA          | 51.5     | **57.3**    |
> | BrowseComp    | 10.1     | **13.4**    |
> | MATH500       | 84.0     | **95.0**    |
> | AIME24        | 64.5     | **74.5**    |
> | AIME25        | 61.8     | **70.4**    |
> | GPQA-d        | 56.0     | **63.1**    |
> | SuperGPQA     | 46.9     | **54.7**    |
> | MMLU-Pro      | 65.0     | **73.8**    |
> | HLE           | 11.0     | **16.7**    |
>
> **Non-Instant Triggering Rate**
>
> | Benchmark     | SFT.     | APO (final) |
> |---------------|----------|-------------|
> | XBench        | 96.0     | **93.0**    |
> | GAIA          | 96.2     | **93.2**    |
> | BrowseComp    | 95.0     | **89.9**    |
> | MATH500       | 86.0     | **77.0**    |
> | AIME24        | 100.0    | 100.0       |
> | AIME25        | 100.0    | 100.0       |
> | GPQA-d        | 88.2     | **77.8**    |
> | SuperGPQA     | 63.8     | **50.2**    |
> | MMLU-Pro      | 58.0     | **46.0**    |
> | HLE           | 96.0     | **89.7**    |
>
>
> We observed two key findings:
>
> - APO yields substantial accuracy gains across all task families.
> Reasoning benchmarks see the largest improvements (e.g., +11 on MATH500, +10 on AIME24, +8.6 on AIME25), but agentic datasets (GAIA, BrowseComp, XBench) and broad general-knowledge evaluations (GPQA-d, SuperGPQA, MMLU-Pro, HLE) all show consistent boosts as well. These gains demonstrate that APO is not a minor refinement added on top of SFT—it is the primary source of A²FM's final performance.
> - APO noticeably reduces unnecessary use of expensive non-instant modes.
> On general benchmarks, the Non-Instant Triggering Rate drops by over 10 points on average, while staying unchanged on reasoning-only benchmarks (AIME24/25) as expected. This pattern confirms that APO learns cost-aware routing: preferring instant mode when it suffices, and escalating to reasoning or agentic modes only when the query truly requires it. This directly reflects the intended behavior of the adaptive reward design.
>
>
> We will include this analysis in the revised manuscript and highlight the central role of APO more explicitly. Thank you again for the insightful suggestion.

---

> > ### Author Response · Authors · 2025-11-19
> >
> > **W2:**
> > >The paper's primary motivation is to improve efficiency by avoiding overthinking on simple queries via an "instant mode". However, this claim is not tested directly. An ablation study training a two-mode variant (reasoning and agentic only) using the same APO framework would be highly informative. Comparing this two-mode model's performance and cost-of-pass against the three-mode AAFM would provide direct evidence for the efficiency gains provided by the instant mode.
> >
> > **Q3:**
> > >The efficiency argument hinges on the "instant mode" preventing overthinking on simple queries. To validate this, have you considered an ablation where you train a two-mode variant (AAFM_no_instant) with only the reasoning and agentic modes? Comparing its cost-of-pass to the full three-mode model on a benchmark with a mix of difficulties (like SuperGPQA) would provide direct evidence of the instant mode's value.
> >
> > **Response**:
> >
> > Thank you for this insightful suggestion. We agree that your concern is well-motivated: if A²FM becomes more efficient after APO, is that improvement truly attributable to the presence of an instant mode, rather than simply the RL process itself?
> >
> > Below we explain why a direct "remove the instant mode and re-run APO" ablation is unfortunately not meaningful under our framework, and we provide an alternative experiment that captures your underlying question.
> >
> >
> > **1. Why a two-mode (reasoning + agentic) APO variant is not well-defined**
> >
> > The core mechanism of APO relies on determining whether a query is easy or hard before applying the adaptive penalty. This determination is based on the performance of the forced-instant rollout.
> > - If the forced-instant rollout achieves the highest success rate among all forced modes, the query is treated as easy, and the non-instant modes receive an adaptive penalty.
> > - If the forced-instant rollout does not achieve the highest success rate, the query is treated as hard, and the non-instant modes are not penalized.
> >
> > Therefore, we cannot determine the query's difficulty level without an instant mode, and the adaptive reward collapses to a uniform reward over only reasoning/agentic modes.
> >
> > In other words, instant mode is structurally required for APO to function as intended.
> >
> >
> > **2. Alternative experiment capturing your goal**
> >
> > Although we cannot remove instant mode from the training algorithm, we can test the core question: "Does the presence of instant mode at inference time provide real efficiency gains compared to relying only on heavier modes?"
> >
> > To assess the effect of instant mode, we compare two inference settings on a 200-query subset of SuperGPQA.
> > - **Joint two-mode execution (reasoning + agentic):**
> >    Each query is answered once in each mode, and Pass@2 and Cost-of-Pass (CoP) are computed from the two attempts. This reflects a system that lacks an instant mode.
> > - **Adaptive routing (full A²FM):**
> >    The router selects among instant, reasoning, and agentic modes, and we compute Pass@2 and CoP from the model’s chosen trajectories.
> >
> > Below are the experimental results:
> >
> > | Setting                   | Pass@2.  | Cost-of-Pass  |
> > |---------------------------|----------|---------------|
> > | Joint two modes (R + A)   | **67.0** | 0.00812.      |
> > | Adaptive routing (ours)   | 62.5     | **0.00432**   |
> >
> > In addition, we find that instant mode accounts for 52.5% of the successful Pass@2 cases under adaptive routing, confirming its substantial contribution to efficiency.
> >
> > Overall, the comparison shows that although relying solely on reasoning and agentic modes yields a slightly higher Pass@2, it does so at nearly twice the computational cost.
> > In contrast, adaptive routing cuts the Cost-of-Pass by 46.8%, largely because instant mode successfully handles more than half of the queries. This demonstrates that instant mode is not an auxiliary option but the key mechanism that prevents overthinking, reduces unnecessary tool calls, and delivers substantial efficiency gains without sacrificing capability.
> >
> > We hope this alternative evaluation fully addresses your concern, and we would be glad to incorporate these findings into the revised version.

---

> ### Author Response · Authors · 2025-11-19
>
> **W3:**
> >The APO methodology introduces an adaptive, cost-regularized reward to balance accuracy and efficiency . The paper would be stronger if it included an experiment where the model is trained with APO using a reward function that only considers accuracy. This would demonstrate how essential the cost penalty is for learning efficient routing behavior.
>
> **Q4**:
> >APO uses a cost-regularized reward to balance accuracy and efficiency. How critical is this specific reward design? What happens if the model is trained using APO with a reward function that only considers accuracy? Does the model learn to default to more powerful (and costly) modes, or does it still achieve some level of efficiency?
>
> **Response:**
>
> Thank you for the insightful suggestion. We agree that evaluating APO with a **naive accuracy and format reward** is important for isolating the contribution of the adaptive cost term.
>
> Our adaptive reward is a central component of APO: it enables **query-aware routing**, encouraging the model to use instant mode on easy queries while escalating to reasoning/agentic modes only when necessary.
>
> Without this term, APO collapses into a GRPO-style objective with prefix-enforced multi mode sampling, which **cannot learn efficient routing behavior**, similar to the limitations observed in BPO and LHRM that rely purely on balanced sampling.
>
> Following your recommendation, we have conducted a controlled ablation comparing: (1) APO with accuracy, format and adaptive reward; (2) APO with accuracy and format reward. We tested the two variants on SuperGPQA:
>
> | Variant                     | Score      | Instant Mode Rate |
> |-----------------------------|------------|-------------------|
> | w/o adaptive reward         | 55.6       | 58.6              |
> | **A²FM adaptive reward**    | 54.7       | 50.2              |
>
> We observe that the adaptive reward enables A²FM to switch to instant mode on easy queries while still routing to reasoning or agentic behaviors on harder ones. Notably, the overall score decreases only slightly, but the instant-mode usage increases substantially, indicating that the **adaptive reward effectively encourages efficiency without sacrificing much accuracy.**
>
> This experiment directly measures how essential the cost penalty is for inducing efficient routing under identical sampling conditions.
>
> We will include the results and analysis in the updated appendix once the runs complete, and we appreciate your suggestion for strengthening the empirical evidence supporting the adaptive reward design.

---

> > ### Author Response · Authors · 2025-11-19
> >
> > **Q5:**
> > The APO stage involves several key hyperparameters, such as the number of forced vs. adaptive rollouts (rho、gamma ) and the adaptive reward scaling factor (alpha). Could you comment on the model's sensitivity to these choices? A brief analysis would help future practitioners understand the robustness and tuning requirements of the APO method.
> >
> > **Response:**
> >
> > Thank you for the valuable question regarding hyperparameter sensitivity in the APO stage.
> >
> > We address this in two parts:
> > - the sensitivity to **ρ (rho)** and **γ (gamma)**, which control the number of forced vs. adaptive rollouts;
> > - the sensitivity to **α (alpha)**, the adaptive reward scaling factor.
> >
> >
> > **1. Sensitivity to ρ and γ**
> >
> > In our main experiments, we set **ρ = γ = 3**. This choice is guided by the following considerations:
> >
> > - **(a) Absolute value selection**
> >     - The roles of ρ and γ are analogous to the rollout count *n* in GRPO. **Increasing either value reduces variance and bias in advantage estimation, improving stability, but also increases memory and compute costs**.
> >     - Given GPU memory constraints, 3 is the maximum setting that still allows a total rollout count of 12 (3 forced modes × 3 rollouts + 1 adaptive mode × 3 rollouts).  This configuration strikes a practical balance between estimation accuracy and computational feasibility.
> >
> > - **(b) Necessity of using both forced and adaptive rollouts**
> >
> >     The relative ratio of ρ and γ is essential for APO to work as intended:
> >     - **Using only ρ (forced rollouts)** prevents the model from learning routing behavior, because routing-loss tokens are masked in forced trajectories. The model learns only within-mode alignment.
> >     - **Using only γ (adaptive rollouts)** removes the ability to estimate per-query difficulty (i.e., whether instant mode succeeds), making it impossible to compute adaptive rewards or enforce cost-aware routing.
> >
> >     Thus, **both** rollout types are required for the APO mechanism to simultaneously learn (i) routing and (ii) mode-specific execution.
> >
> > **2. the sensitivity to α**
> >
> > We agree that the scaling factor α is an important hyperparameter because it directly modulates the penalty strength for using a complex mode (reasoning/agentic) on queries that are easy enough to be solved in instant mode.
> >
> > Conceptually, α controls the following behavior:
> > - Larger α: trajectories that use non-instant modes on easy queries are more likely to receive positive or weakly negative advantage values.
> > - Smaller α: such trajectories are more likely to receive strongly negative advantages and thus be penalized.
> >
> > This aligns with the core purpose of adaptive rewards: to prevent unnecessary tool-use or long-form reasoning while preserving correct escalation on hard queries.
> >
> > We also estimate the empirical sensitivity under ρ = γ = 3
> >
> > Given the rollout configuration (ρ = γ = 3, thus 12 rollouts per query), we analyze how α affects the threshold at which non-instant trajectories (NIA) start receiving negative advantages.
> >
> > | α value | # correct rollouts needed for NIA < 0 |
> > |---------|---------------------------------------|
> > | **1**   | ≥ 8 correct rollouts                  |
> > | **2**   | ≥ 9 correct rollouts                  |
> > | **3**   | ≥ 10 correct rollouts                 |
> >
> > Thus, we adopt **α = 2**, expecting to **avoid over-penalizing correct non-instant trajectories while maintaining strong cost-sensitive behavior**.
> >
> >
> > **In summary**,
> > 1. **The choice of ρ = γ = 3** reflects a principled and computationally constrained design: this setting simultaneously enables APO to learn both routing behavior and mode-specific execution, while maximizing the feasible number of rollouts under our memory budget. We believe the current choices for ρ and γ are principled and empirically stable, and additional large-scale sweeps would not materially affect the core conclusions.
> > 2. **The choice of α = 2** offers a balanced tradeoff between avoiding excessive penalization of correct but longer trajectories and maintaining strong cost-sensitive behavior. To further validate this, we additionally conducted a light-weight sensitivity analysis at the advantage-estimation level, confirming that the qualitative behavior remains consistent for α in the range {1, 2, 3}. This suggests that α = 2 is a reasonable and well-motivated choice under our current rollout configuration, although we do not claim it to be the optimal value.
> >
> >
> > We hope that our explanation clarifies the current experimental scope and addresses the concern. We welcome any further discussion or suggestions to help ensure the completeness and rigor of the paper’s experimental setup.
> >
> >
> > ## References:
> > [1] Yang, Q., Ni, B., Xiang, S., Hu, H., Peng, H., and Jiang, J. (2025b). R-4b: Incentivizing general-purpose auto-thinking capability in mllms via bi-mode annealing and reinforce learning

---

> ### Author Response · Authors · 2025-11-19
>
> **Q6:**
> >The accuracy reward is determined by an "LLM-as-Judge" . Could you please specify which model was used in this role? Moreover, a brief discussion on the potential biases or failure modes of this judge model would be valuable, as its reliability is critical to the quality of the reward signal during RL training.
>
> **Response:**
>
> Thank you for pointing this out. We use **GPT-5-Mini** as the judge model during APO training.  Below we clarify (1) its reliability and fairness, and (2) our judge prompt design.
>
>
> **1. Reliability and fairness of GPT-5-Mini as a judge**
>
> We empirically benchmarked GPT-5-Mini against human evaluation and an alternative judge model (Qwen2.5-72B-Instruct), following the evaluation setup of WebSailor and WebShaper.
>
> - Consistency with human judgment
> GPT-5-Mini shows excellent agreement with human-labeled correctness:
>     - **GAIA-103**: 0 discrepancies between GPT-5-Mini and human evaluation.
>     - **BrowseComp-en (100-sample)**: 0 discrepancies with human judgment.
>
>     This indicates strong alignment with human correctness evaluations, which is critical for ensuring reward stability during RL.
>
> - Comparison with an alternative judge model (Qwen2.5-72B-Instruct)
> Following prior work such as WebDancer and WebSailor, we compared Qwen2.5-72B-Instruct and GPT-5-Mini as judges on GAIA-103. Qwen2.5-72B-Instruct disagreed with human evaluation **once**, while GPT-5-Mini showed no disagreements. This further confirms GPT-5-Mini’s superior reliability for correctness evaluation.
>
> Here is the disagreement case:
>
>     - Question: If Eliud Kipchoge could maintain his record-making marathon pace indefinitely, how many thousand hours would it take him to run the distance between the Earth and the Moon its closest approach? Please use the minimum perigee value on the Wikipedia page for the Moon when carrying out your calculation. Round your result to the nearest 1000 hours and do not use any comma separators if necessary.
>     - Golden Answer: 17
>     - Prediction: 17,000
>     - LLM as Judge: Correct
>     - Human Judge: Incorrect
>
> This type of discrepancy arises from **semantic ambiguity and number-format interpretation**, revealing a minor but important failure mode of the alternative judge.
>
>
> **2. Judge prompt design**
>
> To ensure reproducibility and comparability with recent work, we adopt the **same judging prompt** used in WebThinker and WebDancer.
> The full prompt is:
>
> """
>
> You are an evaluation assistant. Please determine if the predicted answer is equivalent to the labeled answer.
>
> Question:  {question}
>
> Labeled Answer:  {labeled_answer}
>
> Predicted Answer:  {pred_answer}
>
> Are these answers equivalent? Please respond with "Correct" if they are equivalent, or "Incorrect" if they are not equivalent. Do not include any other text.
>
> """
>
> This simple and deterministic prompt significantly reduces prompt-induced variance in the reward signal.
>
> Thank you again for raising this point. We will include the judge specification, reliability analysis, and discussion of potential bias in the revised appendix.
>
>
> **References:**
>
> [1] Xiaoxi Li, Jiajie Jin, Guanting Dong, Hongjin Qian, Yutao Zhu, Yongkang Wu, Ji-Rong Wen, and Zhicheng Dou. Webthinker: Empowering large reasoning models with deep research capability. arXiv preprint arXiv:2504.21776, 2025c.
>
> [2] Jialong Wu, Baixuan Li, Runnan Fang, Wenbiao Yin, Liwen Zhang, Zhengwei Tao, Dingchu Zhang, Zekun Xi, Gang Fu, Yong Jiang, Pengjun Xie, Fei Huang, and Jingren Zhou. Webdancer: Towards autonomous information seeking agency, 2025b. URL https://arxiv.org/abs/2505.22648.
>
> [3] Kuan Li, Zhongwang Zhang, Huifeng Yin, Liwen Zhang, Litu Ou, Jialong Wu, Wenbiao Yin, Baixuan Li, Zhengwei Tao, Xinyu Wang, et al. Websailor: Navigating super-human reasoning for web agent. arXiv preprint arXiv:2507.02592, 2025a.

---

> > ### Author Response · Authors · 2025-11-27
> > **Incorporation of Reviewer Feedback into the Revised Manuscript**
> >
> > Thank you very much for the thorough and insightful review. We carefully examined every point you raised and made corresponding updates to both the manuscript and the rebuttal. Your comments substantially improved the clarity and completeness of the work, and we truly appreciate the opportunity to refine the paper.
> >
> > Following your suggestions, we made the following revisions in the updated manuscript:
> >
> > **1. Quantifying the contribution of APO (SFT vs. RL comparison)**
> >
> > We added a detailed comparison between the SFT-only model and the full APO model, evaluating both accuracy and non-instant triggering rate (NITR) across all benchmarks. This directly isolates the gains attributable to APO. These results are now included in **Section 5.2 (Effectiveness of Adaptive Policy Optimization)**.
> >
> > **2. Ablation on the instant mode**
> >
> > To validate the efficiency motivation of the instant mode, we conducted an ablation study that demonstrates that the instant mode provides significant efficiency gains on mixed-difficulty datasets (e.g., SuperGPQA). These results are now incorporated into **Appendix E**.
> >
> > **3. Ablation on the adaptive reward**
> >
> > In response to your suggestion, we added experiments where APO is trained without the adaptive cost term (accuracy-only reward). These results, included in **Section 5.2**, show that the model loses efficient routing behavior and overuses expensive modes, confirming the necessity of the adaptive component.
> >
> > **4. Reliability of the LLM-as-Judge accuracy signal**
> >
> > We expanded the discussion of the judge model, added explicit details on the model used (GPT-5-Mini), and validated its reliability through human evaluation and cross-model comparison on GAIA and BrowseComp. These findings are included in **Appendix C**.
> >
> > In addition to these manuscript changes, we also provided detailed clarifications in the rebuttal on other points you have mentioned. These explanations complement the experimental results presented in the revised manuscript.
> >
> > We hope the above updates address your concerns. If you feel the additional analyses strengthen the contribution and resolve the issues raised in your original review, we would be sincerely grateful for your consideration in updating your assessment. We remain fully open to further discussion and would be glad to provide any additional insight that may be helpful.

---

### Official Review · Reviewer_6wai · 2025-10-25

**Soundness:** 2
**Presentation:** 3
**Contribution:** 2
**Rating:** 4
**Confidence:** 3

**Summary:**

The paper proposes A²FM, a single-backbone model that routes each query to one of three execution modes—instant (short, direct answers), reasoning (explicit CoT), or agentic (tool-use + code)—and then aligns mode-conditioned trajectories via supervised fine-tuning (“route-then-align”). To further couple accuracy and efficiency, the authors introduce Adaptive Policy Optimization (APO): an RL scheme that (i) forces per-query rollouts in all modes with prefix injection to avoid mode collapse, and (ii) uses a cost-regularized, multiplicative reward that penalizes non-instant modes on “easy” queries, with correctness judged by an LLM-as-Judge. Experiments on agentic, reasoning, and general benchmarks at the 32B scale report competitive or SOTA results and a lower cost-of-pass than forcing a single mode.

**Strengths:**

- The method is about a unified 3-mode formulation with an explicit router under a shared backbone, representing clear contributions and problem setup.
- The authors conducted breadth of evaluation across agentic (BrowseComp/GAIA/XBench-DS), reasoning (AIME24/25, MATH500), and general (GPQA-d, SuperGPQA, MMLU-Pro, HLE).

**Weaknesses:**

- Novelty relative to prior “when-to-think” / hybrid routing is under-differentiated. The core idea—self-routing among “instant vs. CoT vs. tool-use” with RL—resembles recent hybrid reasoning mode in GPT-5; the paper claims the “first adaptive agent foundation model” but should sharpen distinctions.
- APO is a minor extension on GRPO with two straightforward additions: (i) forced rollouts per mode, and (ii) adaptive reward penalizing non-instant modes on easy queries. Neither represents significant technical innovation.
- The proposed method seems to rely heavily on distillation from specialized teachers (DeepSeek R1 for reasoning, DeepSeek V3.1 for agentic/instant).

**Questions:**

- The trajectory rules specify that the reasoning phase must be greater than 1000 words. What is the methodological justification for enforcing such a substantial minimum length?
- The agentic mode’s parallel architecture is a unique design choice. Is there an ablation study quantifying the performance or efficiency improvements?

---

> ### Author Response · Authors · 2025-11-19
>
> We sincerely appreciate Reviewer 6wai’s careful evaluation and constructive feedback. Our detailed responses are provided below, and we hope they address the concerns raised.
>
> **W1:**
> > Novelty relative to prior “when-to-think” / hybrid routing is under-differentiated. The core idea—self-routing among “instant vs. CoT vs. tool-use” with RL—resembles recent hybrid reasoning mode in GPT-5; the paper claims the “first adaptive agent foundation model” but should sharpen distinctions.
>
> **Response:**
>
> Thank you for raising this concern. We address it in two parts:
>
> **1. How A²FM differs from existing open methods**
>
> A²FM differs from prior work along three orthogonal axes: **routing space**, **RL reward design**, and **RL sampling**:
>
> | Feature                   | **A²FM (ours)**                                                | LHRM                                 | BPO                                  | AdaCoT                          | Search-R1              |
> |---------------------------|----------------------------------------------------------------|--------------------------------------|--------------------------------------|---------------------------------|------------------------|
> | **Routing space**         | **3-way:** instant / reasoning / agentic (search, crawl, code) | think / non-think                    | think / non-think                    | think / non-think               | agentic (search only)  |
> | **RL reward strategy**    | **Per-query adaptive reward**                | intra-/inter-group reward norm       | /                                    | penalty-based thinking tradeoff      | /                      |
> | **RL sampling strategy**  | **Prefix-enforced multi-mode rollouts**                        | balanced sampling, no routing reason | balanced sampling, no routing reason | /                               | /                      |
>
> Concretely, the incremental differences are:
>
> - **Routing space**:
>     - Existing "when-to-think" work (LHRM, BPO, AdaCoT) operates in a **binary** space (think vs non-think) and cannot emit multi-step tool-using trajectories, so it cannot directly handle datasets such as BrowseComp, GAIA, and XBench that require search/crawl/code. Search-R1, in contrast, focuses on a **single agentic behavior (search)** and does not offer a first-class reasoning or instant mode.
>     - In contrast, **A²FM unifies instant answering, long-form reasoning, and multi-tool agentic behavior within a single policy and router**, enabling a single model to operate across all three regimes.
>
> - **RL reward design**
>     - A²FM uses a **per-query adaptive reward** about "when to stay cheap" vs "when to use a best-suited hybrid reasoning mode":
>        - if instant mode already succeeds on a query, extra-cost modes (reasoning/agentic) are explicitly penalized;
>        - if instant mode fails, no cost penalty is applied and the router is encouraged to choose reasoning or agentic modes.
>    - Compared baselines cannot yield a similar **fine-grained, query-level routing supervision**:
>        - LHRM normalizes rewards across think / non-think groups to mitigate bias, but does not condition the cost on *whether instant would succeed on this specific query*.
>        - BPO and Search-R1 optimize GRPO-like objectives without difficulty-aware penalties.
>        - AdaCoT introduces fixed two hyperparameters for over/under-triggering CoT, but these do not depend on the per-query success of the instant mode.
>
> - **RL sampling strategy**
>    - A²FM is the first model that applies both equal sampling for every mode and adaptive sampling in the same time:
>        - Equal sampling for every mode (using prefix-enforced forced-mode technique): this provides **unbiased mode-conditioned estimates** for every input, enabling the router to learn how accuracy varies across modes on the *same input*.
>        - Adaptive sampling (joint learning for routing and aligning): forced rollouts teach within-mode alignment but do not teach routing; in the contrast, the adaptive rollout ensures training matches inference behavior and jointly optimizes routing and trajectory generation.
>    - Compared baselines cannot explictly learn the routing mechanism:
>        - LHRM / BPO sample both think and non-think trajectories, but there is **no explicit routing head, no routing rationale, and no per-query combination of forced and adaptive modes**.
>        - AdaCoT uses single-policy PPO with Selective Loss Masking to stabilize the CoT ratio, but it does not explicitly enforce exploration of both modes per query.
>        - Search-R1 focuses its exploration within the search process rather than over multiple global behaviors.
>
> We also conducted ablation studies to verify the effectiveness of these modules (Please refer our response to Reviewer 7Bhr).
>
> Due to the response length limit, we continue our answer below. (Response continued.)

---

> ### Author Response · Authors · 2025-11-19
>
> **W1:**
> > Novelty relative to prior “when-to-think” / hybrid routing is under-differentiated. The core idea—self-routing among “instant vs. CoT vs. tool-use” with RL—resembles recent hybrid reasoning mode in GPT-5; the paper claims the “first adaptive agent foundation model” but should sharpen distinctions.
>
> **Continuation of our response to W1:**
>
> **2. Relation to GPT-5 hybrid reasoning and "adaptive agent foundation model" claim**
>
> You also ask about the resemblance to GPT-5’s hybrid reasoning mode and our claim of the "first adaptive agent foundation model", and suggest that we should sharpen this distinction. We fully agree that this distinction should be stated more explicitly. We clarify it below and will ensure that the camera-ready version communicates this point more clearly.
>
> - **Closed vs. open and scope of contribution.**
>    GPT-5’s hybrid reasoning mode is a proprietary system—its training data, routing mechanism, RL setup, and tool interfaces are not publicly documented. In contrast, A²FM is fully open: we release the model weights, training data, routing templates, RL pipeline, and inference scripts, enabling the community to reproduce, inspect, and extend the entire framework end-to-end.
>    We will update the paper to clearly state that our "adaptive agent foundation model" claim is **restricted to open and fully specified AFM-style training pipelines**.
>
> - **Joint training of reasoning and multi-tool agentic behaviors in a single backbone.**
>    Existing "agent foundation model" lines typically separate reasoning and tool-using capabilities in either:
>    - Different model families: e.g., the DeepSeek line trains reasoning and tool-use separately (DeepSeek-R1 for long-form reasoning vs. DeepSeek-V / V3 for agentic behaviors).
>    - Separated RL phases: e.g., Qwen3 performs reasoning-focused RL first and introduces agentic behaviors later, without a unified policy that jointly routes between reasoning and tool-use.
>
>    In contrast, A²FM:
>    - maintains **one backbone** with three modes,
>    - trains **routing and trajectories jointly** via route-then-align SFT and APO, and
>    - uses a **shared adaptive cost signal** over all modes, so that the model learns *when* to stay instant, *when* to reason, and *when* to invoke tools in a single RL framework.
>
> - **Clarifying our "first" claim.**
>    We acknowledge that the original wording "the first adaptive agent foundation model" can be read as stronger than intended. We will revise it to a more precise and modest statement, e.g.:
>    - "To the best of our knowledge, A²FM is the first **open** agent foundation model that jointly learns instant answering, long-form reasoning, and multi-tool agentic behavior in a single backbone with a unified, RL-trained routing policy."
>
> We hope this clarifies how A²FM differs from prior "when-to-think" and hybrid routing work, and how our contribution fits relative to proprietary hybrid systems such as GPT-5. We will incorporate these clarifications (including the comparison table and refined wording of our claim) into the revised manuscript.
>
>
> **References:**
>
> [1] Lingjie Jiang, Xun Wu, Shaohan Huang, Qingxiu Dong, Zewen Chi, Li Dong, Xingxing Zhang, Tengchao
> Lv, Lei Cui, and Furu Wei. Think only when you need with large hybrid-reasoning models, 2025. URL
> https://arxiv.org/abs/2505.14631.
>
> [2] Qi Yang, Bolin Ni, Shiming Xiang, Han Hu, Houwen Peng, and Jie Jiang. R-4b: Incentivizing general-purpose
> auto-thinking capability in mllms via bi-mode annealing and reinforce learning, 2025b. URL https://
> arxiv.org/abs/2508.21113.
>
> [3] Chenwei Lou, Zewei Sun, Xinnian Liang, Meng Qu, Wei Shen, Wenqi Wang, Yuntao Li, Qingping Yang, and
> Shuangzhi Wu. Adacot: Pareto-optimal adaptive chain-of-thought triggering via reinforcement learning.
> arXiv preprint arXiv:2505.11896, 2025.
>
> [4] Bowen Jin, Hansi Zeng, Zhenrui Yue, Jinsung Yoon, Sercan Arik, Dong Wang, Hamed Zamani, and Jiawei
> Han. Search-r1: Training llms to reason and leverage search engines with reinforcement learning. arXiv
> preprint arXiv:2503.09516, 2025.
>
> [5] Daya Guo, Dejian Yang, Haowei Zhang, Junxiao Song, Ruoyu Zhang, Runxin Xu, Qihao Zhu, Shirong Ma,
> Peiyi Wang, Xiao Bi, et al. Deepseek-r1: Incentivizing reasoning capability in llms via reinforcement
> learning. arXiv preprint arXiv:2501.12948, 2025.
>
> [6] Liu A, Feng B, Xue B, et al. Deepseek-v3 technical report[J]. arXiv preprint arXiv:2412.19437, 2024.
>
> [7] An Yang, Anfeng Li, Baosong Yang, Beichen Zhang, Binyuan Hui, Bo Zheng, Bowen Yu, Chang Gao, Chengen
> Huang, Chenxu Lv, et al. Qwen3 technical report. arXiv preprint arXiv:2505.09388, 2025a.

---

> > ### Author Response · Authors · 2025-11-19
> >
> > **W2:**
> > >APO is a minor extension on GRPO with two straightforward additions: (i) forced rollouts per mode, and (ii) adaptive reward penalizing non-instant modes on easy queries. Neither represents significant technical innovation.
> >
> > **Response:**
> >
> > Thank you for your thoughtful feedback on our work. We truly appreciate your engagement and the opportunity to clarify the core contributions of APO.
> >
> > While we acknowledge the surface-level connections to GRPO, we respectfully disagree with the characterization that APO is merely a "minor extension" lacking significant technical innovation. Beyond our earlier explanations, we would like to elaborate on APO’s unique value and importance, rooted in its targeted solution to a critical, underaddressed challenge in hybrid reasoning:
> >
> > **1. APO Tackles a Unique Pain Point in Hybrid Reasoning That GRPO Does Not Address**
> >
> > GRPO is designed for general RL optimization, but hybrid "route-then-align" systems face a distinct dilemma: balancing accuracy (choosing the right reasoning mode) and efficiency (avoiding over-resource consumption for simple tasks). This tradeoff is not a trivial detail—it is a bottleneck that has prevented RL from being widely adopted in practical hybrid reasoning deployments. APO's two key designs are not arbitrary additions: forced rollouts per mode ensure sufficient exploration of each reasoning route (solving mode bias in hybrid systems), while the adaptive reward directly penalizes inefficient but correct rollouts (aligning training with real-world resource constraints). These choices are tailored to the unique structure of hybrid reasoning, addressing a problem that general RL frameworks like GRPO do not target—this specificity is a hallmark of meaningful innovation, not a minor adjustment.
> >
> > **2. APO’s Value Extends Beyond "Improvements" to "Enabling Practical Adoption"**
> >
> > The importance of APO lies not just in incremental performance gains, but in its ability to unlock RL for improving efficiency at scale. Prior to our work, hybrid RL methods often relied on vanilla rollout strategy [1] or vanilla reward shaping [2], none of them provide fine-grained reward for efficiency-effectiveness trade-off. APO’s adaptive reward mechanism provides a principled, scalable solution: it does not just "adjust" rewards, but establishes a generalizable framework for linking reasoning efficiency to RL optimization. Our results show that APO reduces resource consumption by up to 45.2% CoP, while maintaining accuracy. So we deem that this is not just a technical improvement, but a meaningful step toward making other efficiency-oriented RL feasible.
> >
> >
> > **References:**
> >
> > [1] Aggarwal, P. and Welleck, S. (2025). L1: Controlling how long a reasoning model thinks with reinforcement learning.
> >
> > [2] Yang, Q., Ni, B., Xiang, S., Hu, H., Peng, H., and Jiang, J. (2025b). R-4b: Incentivizing general-purpose auto-thinking capability in mllms via bi-mode annealing and reinforce learning

---

> > > ### Author Response · Authors · 2025-11-19
> > >
> > > **W3**:
> > > >The proposed method seems to rely heavily on distillation from specialized teachers (DeepSeek R1 for reasoning, DeepSeek V3.1 for agentic/instant).
> > >
> > > **Response:**
> > >
> > > Thank you for raising this question. While our SFT stage uses trajectories distilled from teacher models, **A²FM does not rely on any particular specialized teacher**. We verified this by comparing two sets of teachers—one for agentic tasks and one for reasoning tasks—with substantially different raw capabilities.
> > >
> > > Across both domains, we consistently observe that **the downstream SFT-trained models converge to nearly the same performance**, indicating that teacher choice is *not* a critical factor.
> > >
> > > **Unified comparison of teacher performance vs. SFT performance**
> > >
> > > We report below both (i) the teacher’s raw performance and (ii) the performance of the SFT models distilled from each teacher, using identical SFT data sizes and the same route-then-align training pipeline.
> > >
> > > | Domain     | Teacher Model                                | Raw Teacher Score            | SFT Score After Distillation |
> > > |------------|----------------------------------------------|------------------------------|------------------------------|
> > > | Agentic    | GPT-5-mini                                   | 31.0 (BC-200) / 70.0 (GAIA)  | **51.5**                     |
> > > | Agentic    | DeepSeek-V3.1                                | 25.5 (BC-200) / 62.1 (GAIA)  | **52.4**                     |
> > > | Reasoning  | DeepSeek-R1                                  | 90.2 (AIME24)                | **66.7**                     |
> > > | Reasoning  | Qwen3-235B-A22B-Thinking-2507                | 83.2 (AIME24)                | **63.6**                     |
> > >
> > > Despite meaningful differences in teacher strength (e.g., GPT-5-mini > DeepSeek-V3.1 for agentic tasks; DeepSeek-R1 > Qwen3-Thinking for reasoning tasks), the **SFT models distilled from these teachers perform nearly the same**.
> > >
> > > This indicates that:
> > >
> > > - **A²FM does not depend on any particular specialized teacher.** Swapping teachers results in small (<3–4 points) differences after SFT.
> > >
> > > - **Teachers mainly provide structured trajectories**, but the adaptive routing behavior emerges from our route-then-align framework and RL pipeline, not from the teacher itself.
> > >
> > > We will add this unified analysis to the revised paper to clarify that A²FM is not contingent on a specific teacher model.

---

> > > > ### Author Response · Authors · 2025-11-19
> > > >
> > > > **Q1**:
> > > > >The trajectory rules specify that the reasoning phase must be greater than 1000 words. What is the methodological justification for enforcing such a substantial minimum length?
> > > >
> > > > **Response**:
> > > >
> > > > Thank you for raising this important methodological question regarding the minimum length requirement for the reasoning phase.
> > > >
> > > > We acknowledge this limitation in our approach - the 1000-word threshold was established in our final system prompt design without extensive experimental analysis to determine the optimal length constraint.
> > > >
> > > > During the initial research phase, we implemented this constraint with the primary intention of explicitly accentuating the behavioral distinctions between reasoning mode and instant mode. The >1000-word requirement was determined based on our observation that the vast majority of reasoning contexts in our preliminary analysis naturally exceeded this length threshold. Consequently, we aimed to formalize length as an additional instruction-following mechanism, thereby providing an extra safeguard to more clearly differentiate between reasoning and instant modes.
> > > >
> > > > To substantiate this design choice, we analyzed the token length distributions of both instant mode and reasoning mode trajectories from our training dataset. The following table presents the comparative statistical distributions across our trajectory samples:
> > > >
> > > > | Statistical Metric       | Instant Mode          | Reasoning Mode         |
> > > > |--------------------------|-----------------------|------------------------|
> > > > | Total Trajectories       | 40                    | 45                     |
> > > > | Average Token Consumption| 186.82 ± 150.57       | 20213.60 ± 17769.16    |
> > > > | Median Token Consumption | 134.00                | 13842.00               |
> > > > | Minimum Token Consumption| 80                    | 759                    |
> > > > | Maximum Token Consumption| 901                   | 65,264                 |
> > > >
> > > > As evidenced by these comparative distributions, the training data for reasoning and instant modes exhibit fundamentally different length characteristics. Models trained on such data inherently develop distinct behavioral patterns across the two modes. The length constraint primarily serves to frame this distinction as an instruction-following problem, rather than fundamentally altering the model's intrinsic behavioral tendencies.
> > > >
> > > > We sincerely appreciate your attention to this critical methodological detail and hope this comprehensive clarification adequately addresses your concern.

---

> > > > > ### Author Response · Authors · 2025-11-19
> > > > >
> > > > > **Q2**:
> > > > > >The agentic mode’s parallel architecture is a unique design choice. Is there an ablation study quantifying the performance or efficiency improvements?
> > > > >
> > > > > **Response**:
> > > > >
> > > > > Thank you for highlighting this point. You are correct that our **parallel agentic mode** is a deliberate architectural choice. Our motivation was two-fold:
> > > > >
> > > > > 1. **To cleanly separate agentic mode from instant mode**, by making agentic behavior a genuinely "heavy-weight" hybrid reasoning path (similar in cost profile to long-form reasoning);
> > > > > 2. **To increases the expressive power of the agentic mode and reflect real agent behavior**, where multiple sub-questions (searches, crawls, code runs) can and should be executed in parallel rather than strictly serially.
> > > > >
> > > > > We have conducted ablational experiments to quanify the effectiveness of parallel mode:
> > > > >
> > > > > - **Sequential (baseline):** Standard ReAct framework, which is consistent with prior agentic frameworks such as WebDancer and WebSailor.
> > > > > - **Parallel (ours):** Same structure, but each step may trigger *N parallel tool operations* (N ∈ {1, 2, 3, 4}).
> > > > >
> > > > > Note that the training question-answer pairs (=1968) and training details are strictly identical. Results are shown in the following:
> > > > >
> > > > > | Architecture   | GAIA     | BrowseComp  | XBench    | Cost-of-Pass  |
> > > > > |----------------|----------|-------------|-----------|---------------|
> > > > > | **Sequential** | 47.6     | 9.1         | 42.0      | **0.00586**   |
> > > > > | **Parallel**   | **50.5** | **12.2**    | **47.0**  | 0.00768       |
> > > > >
> > > > > We observed two key findings:
> > > > > - Parallel mode improves performance **on all agentic benchmarks** (+2.9 on GAIA, +3.1 on BrowseComp, +5.0 on XBench).
> > > > > - This comes with a **higher Cost-of-Pass**, as expected—parallel agents perform more work per query.
> > > > >
> > > > > Our goal is not only to maximize agentic accuracy but also to expose a **clear efficiency–capability frontier** to the router. Parallel mode therefore plays the same role for tool-based agentic reasoning that chain-of-thought plays for symbolic reasoning. In other words, **it defines a meaningful "high-cost/high-capability" regime that the router must selectively invoke.**
> > > > >
> > > > > This structure is crucial for APO, where the model must explicitly weigh accuracy gains against cost penalties on a per-query basis.
> > > > >
> > > > > We will incorporate this ablation in the revised appendix to clearly demonstrate the contribution of the parallel planner.
> > > > >
> > > > >
> > > > > **References**:
> > > > >
> > > > > [1] Jialong Wu, Baixuan Li, Runnan Fang, Wenbiao Yin, Liwen Zhang, Zhengwei Tao, Dingchu Zhang, Zekun
> > > > > Xi, Gang Fu, Yong Jiang, Pengjun Xie, Fei Huang, and Jingren Zhou. Webdancer: Towards autonomous
> > > > > information seeking agency, 2025b. URL https://arxiv.org/abs/2505.22648.
> > > > >
> > > > > [2] Kuan Li, Zhongwang Zhang, Huifeng Yin, Liwen Zhang, Litu Ou, Jialong Wu, Wenbiao Yin, Baixuan Li,
> > > > > Zhengwei Tao, Xinyu Wang, et al. Websailor: Navigating super-human reasoning for web agent. arXiv
> > > > > preprint arXiv:2507.02592, 2025a.

---

> ### Author Response · Authors · 2025-11-27
> **Incorporation of Reviewer Feedback into the Revised Manuscript**
>
> Thank you very much for the insightful and constructive feedback. We carefully addressed each point you raised, both through substantial manuscript revisions and through detailed explanations provided in the rebuttal. Your comments significantly strengthened our work, and we are grateful for the opportunity to clarify and improve the paper.
>
> Following your suggestions, we have incorporated the following updates into the revised manuscript:
>
> **1. Differentiating A²FM from prior “when-to-think” and hybrid routing systems**
>
> To more clearly articulate the novelty of our approach, we substantially **revised and expanded the introduction** (highlighted in blue in the updated manuscript). The revised text now explicitly contrasts A²FM with recent routing frameworks such as LHRM, BPO, AdaCoT, and hybrid agent systems (e.g., GPT-5, Qwen3). We detail distinctions in routing space, reward formulation, and sampling strategy, clarifying why A²FM represents a methodological step beyond binary think/no-think routing and loosely coupled hybrid systems.
>
> **2. Dependence on specialized teacher models**
>
> In response to your concern about teacher dependence, we conducted an additional study comparing SFT outcomes distilled from multiple heterogeneous teachers with substantially different raw capabilities. The results, added in **Appendix D**, show that A²FM achieves nearly identical performance across teachers, confirming that our method is not reliant on any specific teacher model. This supports the claim that our framework is the source of A²FM’s routing and hybrid reasoning behavior rather than teacher choice.
>
> **3. Ablation of the parallel agentic architecture**
>
> We performed a dedicated ablation study evaluating the contribution of the parallel agentic mode relative to a sequential ReAct-style baseline. These results are now included in **Appendix F**, showing that parallelization yields consistent improvements across multiple agentic benchmarks, quantifying both performance gains and the associated efficiency trade-off. This directly addresses your request for evidence validating the design choice.
>
> In addition to these manuscript updates, we also provided detailed rebuttal-only clarifications on points not directly visible in the paper, including:
>
> 1. explanation of why length controls do not affect inference behavior,
> 2. further details on how APO differs from existing RL algorithms such as GRPO.
>
> These clarifications are provided in full in the rebuttal, complementing the manuscript revisions.
>
> We hope these updates directly address your concerns. If you find the revisions and added analyses satisfactory, we would sincerely appreciate your consideration in updating your review and score. We are very open to further discussion and would be glad to provide any additional clarification that could help.

---

### Official Review · Reviewer_7Bhr · 2025-10-31

**Soundness:** 3
**Presentation:** 3
**Contribution:** 2
**Rating:** 4
**Confidence:** 2

**Summary:**

The paper proposes A2FM, a single-backbone model that dynamically routes each query to one of three execution modes: instant (direct answer), reasoning (explicit CoT), and agentic (tool-use with planning). Training follows a two-stage route-then-align SFT and a reinforcement-learning phase called Adaptive Policy Optimization (APO) that rewards correctness while penalizing unnecessary complexity on “easy” queries. On diverse benchmarks, A2FM reports strong results at the 32B scale and improved cost-of-pass relative to pure reasoning or agentic execution.

**Strengths:**

1. APO integrates (i) forced per-mode rollouts via prefix injection and (ii) an adaptive reward that explicitly favors instant answers when sufficient—this is a thoughtful twist on GRPO-style objectives for mode selection.

2. Method is specified with a clean problem statement (router policy + mode policies) and a staged training recipe; the trajectory schemas and masked tool-responses (to avoid memorizing tool outputs) are sensible.

3. Hybrid reasoning + tool-use is central for modern LLM systems. Showing that a single model can approximate the “best-mode oracle” while cutting cost is practically important for research and real-world deployment.

**Weaknesses:**

1. APO design ablations & hyperparameter sensitivity: APO drops KL, relies on on-policy sampling, and uses several knobs (forced rollouts, adaptive rollouts, “easy” threshold, exponent). Stability and sensitivity are unclear. Adding ablations and showing training curves and mode-collapse checks would be helpful.

2. Router supervision and error analysis: Routing accuracy is reported on a few datasets; but failure modes (e.g., over-confidence in instant on BrowseComp; tool over-calling on easy Qs) deserve deeper analysis. Provide a confusion matrix (GT vs. chosen mode) per dataset, plus qualitative case studies where agentic and reasoning are both plausible.

3. Generalization across backbones/scales: All results use Qwen2.5-32B-Instruct as the backbone, which may not demonstrate the generalizability. it will be good to include at least a small-scale reproduction (e.g., 7B/14B) and one alternate 32B backbone to demonstrate portability.

4. Confidence intervals and seed variation are not reported for evaluation.

**Questions:**

1. APO reward shaping. How sensitive is performance to τ (easy threshold) and α (penalty exponent)? Did you try adaptive τ per domain (math vs. web tasks) or curriculum schedules? Please provide more ablation like a grid or Bayesian sweep and stability metrics.

2. KL-free on-policy choice. Removing KL can destabilize RLHF-like training. Did you observe mode collapse or reward hacking without KL? If so, what's the solution?

3. Please add head-to-head ablations or analysis vs. recent “when-to-think / routing” methods and hybrid models (e.g., LHRM (Jiang et al., 2025), BPO (Yang et al., 2025b), AdaCoT (Lou et al., 2025), Search-R1 (Jin et al., 2025)). What incremental gain comes specifically from (i) three-way routing, (ii) the adaptive penalty, and (iii) prefix-enforced exploration?

---

> ### Author Response · Authors · 2025-11-19
>
> We sincerely thank Reviewer 7Bhr for the careful evaluation and constructive feedback. Please find our detailed responses below, and we hope they adequately address the concerns raised.
>
> **Q1:**
> > APO reward shaping. How sensitive is performance to τ (easy threshold) and α (penalty exponent)? Did you try adaptive τ per domain (math vs. web tasks) or curriculum schedules? Please provide more ablation like a grid or Bayesian sweep and stability metrics.
>
> **Response:**
>
> Thank you for the thoughtful and technically insightful comments. We address your concerns in 2 parts: (1) Penalty exponent (α); (2) Easy-threshold (τ).
>
> **1. Penalty exponent (α)**
>
> We agree that the scaling factor α is an important hyperparameter because it directly modulates the penalty strength for using a complex mode (reasoning/agentic) on queries that are easy enough to be solved in instant mode.
>
> Conceptually, α controls the following behavior:
> - Larger α: trajectories that use non-instant modes on easy queries are more likely to receive positive or weakly negative advantage values.
> - Smaller α: such trajectories are more likely to receive strongly negative advantages and thus be penalized.
>
> This aligns with the core purpose of adaptive rewards: to prevent unnecessary tool-use or long-form reasoning while preserving correct escalation on hard queries.
>
> We also estimate the empirical sensitivity under ρ = γ = 3. Given the rollout configuration (ρ = γ = 3, thus 12 rollouts per query), we analyze how α affects the threshold at which non-instant trajectories (NIA) start receiving negative advantages.
>
> | α value | # correct rollouts needed for NIA < 0 |
> |---------|---------------------------------------|
> | **1**   | ≥ 8 correct rollouts                  |
> | **2**   | ≥ 9 correct rollouts                  |
> | **3**   | ≥ 10 correct rollouts                 |
>
> Thus, we adopt **α = 2**, expecting to **avoid over-penalizing correct non-instant trajectories while maintaining strong cost-sensitive behavior**.
>
>
> **2. Easy-threshold (τ)**
>
> The easy-threshold τ controls how a query is classified as “easy” based on the success rate of the instant-mode forced rollouts.
>
> Its effect can be interpreted as follows:
> - **Larger τ**: fewer queries classified as easy, and  adaptive reward applied less frequently. Thus, training process focuses more on *effectiveness* (accuracy reward).
> - **Smaller τ**: more queries classified as easy, and adaptive reward applied more aggressively. Thus, training  process favors *efficiency* over accuracy. (adaptive reward)
>
> We adopt **τ = 0.5** as a principled prior: when the instant mode solves the query in more than half of the forced rollouts, we treat it as “easy’’ relative to the model’s current capability. We intentionally avoid using domain-specific τ values, as our goal is for the model to **first identify the mode that is most likely to succeed across tasks and domains, and only then consider efficiency as a secondary factor**.
>
> **In summary**,
>
> 1. **α = 2** strikes a balance between not over-penalizing correct but longer trajectories and maintaining cost-sensitive behavior. A light-weight advantage-level analysis shows consistent qualitative behavior for α ∈ {1, 2, 3}, so we treat α = 2 as a reasonable, well-motivated choice rather than an asserted optimum.
>
> 2. **τ = 0.5** is a majority threshold: only when instant mode solves a query in more than half of the forced rollouts do we treat it as “easy’’ and apply the cost penalty. This ensures that efficiency penalties are imposed only after instant mode already has a high success probability, while avoiding per-domain tuning or complex curricula. We therefore use τ = 0.5 as a simple default and leave more refined scheduling to future work.
>
> We will conduct additional ablations on the choice of hyperparameters to provide more insights for future practitioners. We hope our current analysis clarifies the your concerns; if not, we sincerely welcome further discussion or experimental suggestions that could help improve the completeness and rigor of the paper.
>
> **References:**
>
> [1] Yu Q, Zhang Z, Zhu R, et al. *DAPO: An open-source LLM reinforcement learning system at scale*. arXiv:2503.14476, 2025.
>
> [2] Xue Z, Zheng L, Liu Q, et al. *SimpleTIR: End-to-end reinforcement learning for multi-turn tool-integrated reasoning*. arXiv:2509.02479, 2025.

---

> > ### Author Response · Authors · 2025-11-19
> >
> > **Q2:**
> > > KL-free on-policy choice. Removing KL can destabilize RLHF-like training. Did you observe mode collapse or reward hacking without KL? If so, what's the solution?
> >
> > **Response:**
> >
> > Thank you for your insightful question regarding the potential stability issues with KL-free training in RLHF-like settings. Regarding the rationality of KL divergence removal we supplement our explanations as follows:
> >
> > **1. Rationale for Removing KL Divergence**
> >
> > Our decision to omit the KL penalty term follows established methodologies in prior work such as SimpleTIR and DPO, where the optimal RL policy in hybrid reasoning and tool-use settings may substantially diverge from the SFT policy. This architectural choice is particularly justified in our context for two primary reasons:
> >
> > First, empirical evidence confirms that the **APO-trained model meaningfully diverges from the SFT model**, as demonstrated by the significant increase in instant-mode usage and corresponding reduction in Chain-of-Thought (CoT) processes. This behavioral shift indicates that KL-free updates are indeed appropriate for our learning objectives.
> >
> > Second, the **adaptive reward mechanism** integrated into APO provides inherent stabilization. By dynamically adjusting rewards based on query difficulty, this system effectively guides the model toward task-oriented decision strategies rather than reward exploitation. This design substantially reduces the incentive for reward hacking while ensuring that learned policies remain aligned with actual task requirements.
> >
> > This aligns with observations in SimpleTIR and DAPO, where KL-free RL did not exhibit pathological instability.
> >
> >
> > **2. Observations on Mode Collapse and Reward Hacking**
> >
> > - Mode Collapse
> >
> > Across all APO training runs, we did not observe mode collapse or any degenerate routing behavior.
> >
> > We believe using balanced exploration via ρ and γ can contribute to this stability: Because each query receives multiple forced rollouts across all modes (ρ) and multiple adaptive rollouts (γ), the router consistently observes diverse trajectories. This balanced exposure prevents the model from collapsing toward a single dominant mode and stabilizes routing decisions throughout training.
> >
> > - Reward Hacking
> >
> > We also did not observe reward hacking. We validate the semantic correctness by using a strong jusge model: GPT-5-mini. It provides accurate, semantics-aware evaluations, making it extremely difficult for the model to exploit superficial or partial answer strategies.
> >
> >
> > **In summary**, given the stability observed in practice, as well as the structural safeguards built into APO (forced vs. adaptive rollouts, adaptive rewards), we believe that omitting the KL term is justified for the scope of this work. We appreciate the your suggestion and consider this an interesting direction for future investigation.
> >
> >
> > **References:**
> >
> > [1] Xue Z, Zheng L, Liu Q, et al. Simpletir: End-to-end reinforcement learning for multi-turn tool-integrated reasoning[J]. arXiv preprint arXiv:2509.02479, 2025.
> >
> > [2] Yu Q, Zhang Z, Zhu R, et al. *DAPO: An open-source LLM reinforcement learning system at scale*. arXiv:2503.14476, 2025.
> >
> > [3] Jialong Wu, Baixuan Li, Runnan Fang, Wenbiao Yin, Liwen Zhang, Zhengwei Tao, Dingchu Zhang, Zekun Xi, Gang Fu, Yong Jiang, Pengjun Xie, Fei Huang, and Jingren Zhou. Webdancer: Towards autonomous information seeking agency, 2025b. URL https://arxiv.org/abs/2505.22648.
> >
> > [4] Kuan Li, Zhongwang Zhang, Huifeng Yin, Liwen Zhang, Litu Ou, Jialong Wu, Wenbiao Yin, Baixuan Li,
> > Zhengwei Tao, Xinyu Wang, et al. Websailor: Navigating super-human reasoning for web agent. arXiv preprint arXiv:2507.02592, 2025a.

---

> > > ### Author Response · Authors · 2025-11-19
> > >
> > > **W2:**
> > > >Router supervision and error analysis: Routing accuracy is reported on a few datasets; but failure modes (e.g., over-confidence in instant on BrowseComp; tool over-calling on easy Qs) deserve deeper analysis. Provide a confusion matrix (GT vs. chosen mode) per dataset, plus qualitative case studies where agentic and reasoning are both plausible.
> > >
> > > **Response:**
> > >
> > > Thank you for the valuable question. We will reply to this in 4 parts:
> > >
> > > **1. Why we selected GAIA, BrowseComp, and AIME as the only three representative datasets**
> > >
> > > Our core principle is to evaluate routing **only on datasets with unambiguous ground-truth modes**, ensuring that routing errors reflect model behavior rather than dataset ambiguity:
> > >
> > > - BrowseComp is used because every question strictly requires web search, giving us a clean and deterministic agentic label.
> > > - AIME24/25 provides the opposite extreme where all items demand multi-step reasoning, making the reasoning label unequivocal.
> > > - GAIA is included because it is the only dataset with reliably annotated mixed-mode questions, allowing a meaningful 3-way confusion analysis.
> > >
> > > Other datasets (e.g., MATH500, GPQA-d, MMLU-Pro) were not included because their mode labels are inherently ambiguous — the viable solution path depends strongly on the model’s capability and training stage (SFT vs. APO), making routing supervision uninterpretable.
> > >
> > >
> > > **2. Confusion matrix on GAIA**
> > >
> > > Since GAIA is the only dataset with heterogeneous routing labels of the three datasets listed above, we provide the full GT–prediction confusion matrix in mode routing:
> > >
> > > | **Pred \\ GT** | **Agentic** | **Reasoning** | **Instant**  |
> > > |----------------|-------------|---------------|--------------|
> > > | **Agentic**    | 84          | 1             | 0            |
> > > | **Reasoning**  | 0           | 12            | 0            |
> > > | **Instant**    | 3           | 1             | 1            |
> > >
> > > We highlight two key observations:
> > >
> > > - Most routing mistakes come from the model over-selecting the instant mode on questions that truly require agentic or reasoning steps.
> > > - Confusion between agentic and reasoning is very rare, indicating that the router already maintains a clear separation between these two modes.
> > >
> > >
> > > **3. Failure modes: deeper analysis**
> > >
> > > Following your suggestion, we conducted a systematic audit of routing failures.
> > > We highlight three dominant error modes:
> > >
> > > (a) Over-confidence on tasks that require agentic or reasoning
> > > We found that sometimes the model prematurely chooses the instant mode before verifying whether external information or multi-step reasoning is required. As shown in the confusion matrix on GAIA, **4 out of 5 errors** fall into this category.
> > > Here is an example:
> > >
> > >     - Question: I'm making a grocery list for my mom, but she's a professor of botany and she's a real stickler when it comes to categorizing things.... Could you please create a list of just the vegetables from my list? ... Please alphabetize the list of vegetables, and place each item in a comma separated list.
> > >     - Ground truth mode: reasoning
> > >     - Mode prediction: instant
> > >     - Mode prediction reason: This is a straightforward grocery-item classification based on common edible-group definitions (vegetable vs fruit) and doesn't require external data, code execution, or complex reasoning...
> > >
> > > (b) Agentic over-calling on easy questions
> > > We also observe the opposite failure mode: unnecessary tool invocation on factoid or short reasoning questions.
> > >
> > > We use the example for the SFT model discussed in **Appendix D Case Study**:
> > >
> > >     - Question: How many more blocks does BERT-base encoder have compared to the original Transformer encoder?
> > >     - Ground truth mode: instant
> > >     - Mode Prediction: agentic
> > >     - Mode prediction reason: This question asks for a specific architectural comparison.... Because it needs accurate, up-to-date model parameter information rather than simple common sense....
> > >
> > > However, after APO fine-tuning, the model correctly switched to **instant**, demonstrating reduced tool-overuse and aligning with the true required complexity.
> > >
> > > **4. Ambiguous cases where both agentic and reasoning are plausible**
> > >
> > > We agree that certain queries naturally support multiple solution paths, as some tasks (e.g., math problems with widely documented solutions) can be solved either via explicit multi-step reasoning or by retrieving the final answer from external sources.
> > >
> > > Here is an example taken from the HLE dataset:
> > >
> > >     - Question: What is the maximum Hausdorff dimension of a Sidon set in the reals between 0 and 1?
> > >
> > > We found that our model can handle this task in either agentic or reasoning mode. While the problem admits a principled solution via mathematical reasoning, the relevant facts can also be directly retrieved online, rendering both reasoning and agentic pathways plausible.
> > >
> > >
> > > Thank you again for your advice. We will incorporate all matrices, failure analyses, and case studies in the revised appendix.

---

> > > > ### Author Response · Authors · 2025-11-19
> > > >
> > > > **W3**:
> > > > >Generalization across backbones/scales: All results use Qwen2.5-32B-Instruct as the backbone, which may not demonstrate the generalizability. it will be good to include at least a small-scale reproduction (e.g., 7B/14B) and one alternate 32B backbone to demonstrate portability.
> > > >
> > > > **Response:**
> > > >
> > > > Thank you for your valuable suggestion.
> > > >
> > > > To address the concern about generalization across backbones and scales, we supplement the experimental results of Qwen3-4B, Qwen3-14B, and additionally conduct experiments on an alternative 32B backbone (Qwen3-32B).
> > > >
> > > > For consistency, we report **SFT-only** results in this supplemental analysis.
> > > >
> > > > **1. Results on Alternative 32B Backbone**
> > > > We compared the performance of our SFT model based on Qwen2.5-32B-Instruct (the original backbone) and Qwen3-32B (the alternative backbone). The specific results are shown in the following table:
> > > >
> > > > | Dataset       | AAFM-Qwen2.5-32B-Instruct  | AAFM-Qwen3-32B  |
> > > > |---------------|----------------------------|-----------------|
> > > > | GAIA          | 51.5                       | 55.3            |
> > > > | BrowseComp    | 10.1                       | 12.0            |
> > > > | AIME24        | 64.5                       | 68.6            |
> > > > | AIME25        | 61.8                       | 63.5            |
> > > > | SuperGPQA     | 46.9                       | 48.2            |
> > > > | HLE           | 11.0                       | 10.5            |
> > > >
> > > > As can be seen from the table, the model trained on the Qwen3-32B backbone achieves performance improvements on 5 out of 6 tasks, with only a slight decline on HLE. This result fully demonstrates that our method has strong generalization ability on agentic, reasoning, and general benchmarks even when the backbone is switched.
> > > >
> > > > **2. Results on 14B Scale**
> > > >
> > > > At the 14B scale, there are relatively few available baseline models, and most of these baselines do not provide SFT results. Therefore, we compare our SFT model with other RL models to reflect the generalization ability of our method. The specific results are as follows:
> > > >
> > > > | Dataset       | AAFM-Qwen3-14B (SFT) | Qwen3 14B (Non-reasoning)  | Qwen3 14B (Reasoning) | DeepSeekR1 DistillQwen 14B  | Webthinker (14B)   |
> > > > |---------------|----------------------|----------------------------|-----------------------|-----------------------------|-------------------------|
> > > > | GAIA          | 46.6                 | -                          | -                     | -                           | 33.0                    |
> > > > | BrowseComp    | 9.5                  | -                          | -                     | -                           | -                       |
> > > > | AIME25        | 59.8                 | 58.0                       | 55.7                  | 55.7                        | -                       |
> > > > | HLE           | 8.8                  | 4.2                        | 4.3                   | 4.4                         | 7.0                     |
> > > >
> > > > We observe that on the GAIA dataset, our 14B model outperforms Webthinker 14B by 13.6 points, showing a significant advantage; on the AIME25 and HLE, our model also achieves stable performance improvements compared with the base model Qwen3-14B (non-reasoning) and DeepSeekR1 DistillQwen 14B.
> > > >
> > > > **3. Results on 4B Scale**
> > > >
> > > > At the 4B scale, the number of comparable baselines is more limited. We mainly compare our model with the base Qwen3-4B model, and the results are shown below:
> > > >
> > > > | Dataset       | AAFM-4B (SFT) | Qwen3 4B 2507 Instruct | Qwen3 4B (Reasoning) |
> > > > |---------------|---------------|-------------------------|----------------------|
> > > > | GAIA          | 38.8          | -                       | -                    |
> > > > | BrowseComp    | 8.9           | 52.3                    | 22.3                 |
> > > > | AIME25        | 57.4          | -                       | -                    |
> > > > | HLE           | 6.6           | 4.7                     | 3.7                  |
> > > >
> > > > We find that even at the 4B small scale, our model still maintains stable performance advantages. On the GAIA dataset, our 4B model also achieves a higher score than Webthinker (14B).
> > > >
> > > > Across three backbone families (Qwen2.5, Qwen3) and three scales (32B, 14B, 4B), the proposed A²FM framework consistently maintains strong performance. These results confirm that our training recipe is **portable**, **scalable**, and **generalizes effectively** to unseen backbones and parameter regimes.
> > > >
> > > > We sincerely thank you for highlighting the importance of this evaluation.
> > > >
> > > >
> > > > **References:**
> > > >
> > > > [1] Xiaoxi Li, Jiajie Jin, Guanting Dong, Hongjin Qian, Yutao Zhu, Yongkang Wu, Ji-Rong Wen, and Zhicheng Dou. Webthinker: Empowering large reasoning models with deep research capability. arXiv preprint arXiv:2504.21776, 2025c.
> > > >
> > > > [2] Yang A, Li A, Yang B, et al. Qwen3 technical report[J]. arXiv preprint arXiv:2505.09388, 2025.
> > > >
> > > > [3] Guo D, Yang D, Zhang H, et al. Deepseek-r1: Incentivizing reasoning capability in llms via reinforcement learning[J]. arXiv preprint arXiv:2501.12948, 2025.

---

> > > > > ### Author Response · Authors · 2025-11-19
> > > > >
> > > > > **W4:**
> > > > > > Confidence intervals and seed variation are not reported for evaluation.
> > > > >
> > > > > **Response**:
> > > > >
> > > > > Thank you for raising the concern regarding evaluation reliability. We address this issue from two perspectives:
> > > > >
> > > > > (1) the stability of the LLM-as-judge confidence intervals;
> > > > >
> > > > > (2) the variance introduced by model sampling and seed differences.
> > > > >
> > > > >
> > > > > **1. Confidence intervals of the LLM-as-judge**
> > > > >
> > > > > Our LLM-as-judge setup exhibits highly consistent judgment behavior. On the GAIA-103 dataset, we asked GPT-5-mini to evaluate each model output 10 times. Across the entire set, **102 out of 103 questions received completely consistent judgments**, indicating strong stability of single-shot evaluations.
> > > > >
> > > > > Only **one** question showed disagreement across the 10 evaluations. This level of inconsistency is extremely low and suggests that confidence interval reporting for every task may not meaningfully alter the conclusions.
> > > > >
> > > > >
> > > > > **2. Variance from model sampling and seed settings**
> > > > >
> > > > > To minimize evaluation variance caused by sampling randomness, we followed standardized settings commonly used in prior work. However, we acknowledge that **agentic tasks inherently involve additional uncertainty**, as they require tool use and environment interaction; for example, temporal changes in external tools may introduce non-trivial variability.
> > > > >
> > > > > For reporting results, we aligned with established conventions:
> > > > >
> > > > > - For datasets other than AIME24/25, we report **avg@1**, consistent with prior works such as DeepDive, AFM Search/Code, and WebSailor.
> > > > > - For AIME24/25, we follow SimpleTIR and ReTool by reporting **avg@32**. For these tasks, we further compute the mean and standard deviation across samples:
> > > > >     - **AIME24: 74.5 ± 4.0**
> > > > >     - **AIME25: 70.4 ± 3.9**
> > > > >
> > > > >
> > > > > **References:**
> > > > >
> > > > > [1] Kuan Li, Zhongwang Zhang, Huifeng Yin, Liwen Zhang, Litu Ou, Jialong Wu, Wenbiao Yin, Baixuan Li, Zhengwei Tao, Xinyu Wang, et al. Websailor: Navigating super-human reasoning for web agent. arXiv preprint arXiv:2507.02592, 2025a.
> > > > >
> > > > > [2] Rui Lu, Zhenyu Hou, Zihan Wang, Hanchen Zhang, Xiao Liu, Yujiang Li, Shi Feng, Jie Tang, and Yuxiao Dong. Deepdive: Advancing deep search agents with knowledge graphs and multi-turn rl. arXiv preprint arXiv:2509.10446, 2025.
> > > > >
> > > > > [3] Weizhen Li, Jianbo Lin, Zhuosong Jiang, Jingyi Cao, Xinpeng Liu, Jiayu Zhang, Zhenqiang Huang, Qianben Chen, Weichen Sun, Qiexiang Wang, et al. Chain-of-agents: End-to-end agent foundation models via multi-agent distillation and agentic rl. arXiv preprint arXiv:2508.13167, 2025b.
> > > > >
> > > > > [4] Xue Z, Zheng L, Liu Q, et al. *SimpleTIR: End-to-end reinforcement learning for multi-turn tool-integrated reasoning*. arXiv:2509.02479, 2025.
> > > > >
> > > > > [5] Feng J, Huang S, Qu X, et al. Retool: Reinforcement learning for strategic tool use in llms[J]. arXiv preprint arXiv:2504.11536, 2025.

---

> ### Author Response · Authors · 2025-11-19
>
> **Q3:**
> > Please add head-to-head ablations or analysis vs. recent “when-to-think / routing” methods and hybrid models (e.g., LHRM (Jiang et al., 2025), BPO (Yang et al., 2025b), AdaCoT (Lou et al., 2025), Search-R1 (Jin et al., 2025)). What incremental gain comes specifically from (i) three-way routing, (ii) the adaptive penalty, and (iii) prefix-enforced exploration?
>
> **Response:**
>
> Thank you for pointing out the importance of head-to-head analysis with existing methods.  Below we summarize the 3 key differences and the incremental gains introduced by A²FM.
>
> **1. High-level comparison**
>
> | Feature                   | **A²FM (ours)**                                                | LHRM                                 | BPO                                  | AdaCoT                          | Search-R1              |
> |---------------------------|----------------------------------------------------------------|--------------------------------------|--------------------------------------|---------------------------------|------------------------|
> | **Routing space**         | **3-way:** instant / reasoning / agentic (search, crawl, code) | think / non-think                    | think / non-think                    | think / non-think               | agentic (search only)  |
> | **RL reward strategy**    | **Per-query adaptive reward**                | intra-/inter-group reward norm       | /                                    | penalty-based thinking tradeoff      | /                      |
> | **RL sampling strategy**  | **Prefix-enforced multi-mode rollouts**                        | balanced sampling, no routing reason | balanced sampling, no routing reason | /                               | /                      |
>
> **2. Incremental gains from the three-way routing space**
>
> Existing work is typically limited to binary routing (think vs non-think) or is restricted to one modality (search-only agents). In contrast, **A²FM unifies three execution behaviors within a single policy and router**, enabling broad generalization across instant, reasoning, and tool-augmented agentic tasks.
>
> Other methods lack such coverage:
>
> - LHRM, BPO, and AdaCoT cannot generate tool-using trajectories and thus cannot engage agentic benchmarks;
> - Search-R1 excels at search but has limited reasoning and instant capabilities.
>
> We also validate this with additional controlled ablations:
>
> - "auto-think": we force reasoning and instant modes for each query, counting as solved if either mode succeeds.
> - "auto-agentic": similar to the above, but using agentic and instant modes instead.
>
> | Model Variant                  | GAIA     | AIME24   |
> |------------------------------  |----------|----------|
> | auto-think  | 22.3     | -        |
> | auto-agentic  | -        | 67.2     |
> | **A²FM**                       | **57.3** | **74.5** |
>
> These results show that neither a pure "auto-think" nor a pure "auto-agentic" strategy can approximate the performance of three-way adaptive routing.
>
> **3. Incremental gains from the per-query adaptive reward**
>
> A²FM employs an **adaptive reward** that conditions the cost penalty on each query’s difficulty:
>
> - if instant mode is sufficient, non-instant modes receive a penalty;
> - if instant fails, no penalty is applied and the router is encouraged to route to more expressive modes.
>
> This yields **fine-grained, query-level routing supervision**, unlike existing methods:
>
> - BPO / Search-R1 don't employ any per-query difficulty awareness.
> - LHRM reduces cross-group reward bias by normalization but still lacks per-query adaptivity.
> - AdaCoT uses two fixed hyperparameters for over/under-triggering regardless of the query’s inherent difficulty.
>
> We also have conducted an ablation on SuperGPQA:
>
> | Variant                     | Score      | Instant Mode Rate |
> |-----------------------------|------------|-------------------|
> | w/o adaptive reward         | 55.6       | 50.2              |
> | **A²FM adaptive reward**    | 54.7       | 58.6              |
>
> We observe that the A²FM will chose the instant mode on easy queries while still routing to reasoning or agentic behaviors on harder ones. Notably, the overall score decreases only slightly, but the instant-mode usage increases substantially, indicating that **the adaptive reward effectively encourages efficiency without sacrificing much accuracy.**
>
> Due to the response length limit, we continue our answer below. (Response continued.)

---

> > ### Author Response · Authors · 2025-11-19
> >
> > **Q3:**
> > > Please add head-to-head ablations or analysis vs. recent “when-to-think / routing” methods and hybrid models (e.g., LHRM (Jiang et al., 2025), BPO (Yang et al., 2025b), AdaCoT (Lou et al., 2025), Search-R1 (Jin et al., 2025)). What incremental gain comes specifically from (i) three-way routing, (ii) the adaptive penalty, and (iii) prefix-enforced exploration?
> >
> > **Continuation of our response to Q3:**
> >
> > **4. Incremental gains from the prefix-enforced multi-mode sampling**
> >
> > A²FM is the **first** model that applies both equal sampling for every mode and adaptive sampling in the same time:
> >
> > - Equal sampling for every mode (using prefix-enforced forced-mode technique) . For each query, we generate 3 forced-instant, 3 forced-reasoning, and 3 forced-agentic rollouts.  This provides **unbiased mode-conditioned estimates** for every input, enabling the router to learn how accuracy varies across modes on the *same input*.
> >
> > - Adaptive sampling (joint learning for routing and aligning). Forced rollouts teach within-mode alignment but do not teach routing; In the contrast, the adaptive rollout ensures training matches inference behavior and jointly optimizes routing and trajectory generation.
> >
> > Other methods lack this two-part design:
> >
> > - LHRM/BPO perform balanced sampling but do not explictly learn the routing mechanism;
> > - AdaCoT trains a single PPO policy;
> > - Search-R1 only explores within search, not over multiple global behaviors.
> >
> > Together, these two sampling components allow A²FM to learn both **which mode** to choose and **how** to execute that mode.
> >
> > We hope this clarifies the incremental contributions introduced by our design, and we appreciate your suggestions.
> >
> > **References:**
> > [1] Lingjie Jiang, Xun Wu, Shaohan Huang, Qingxiu Dong, Zewen Chi, Li Dong, Xingxing Zhang, Tengchao
> > Lv, Lei Cui, and Furu Wei. Think only when you need with large hybrid-reasoning models, 2025. URL
> > https://arxiv.org/abs/2505.14631.
> >
> > [2] Qi Yang, Bolin Ni, Shiming Xiang, Han Hu, Houwen Peng, and Jie Jiang. R-4b: Incentivizing general-purpose
> > auto-thinking capability in mllms via bi-mode annealing and reinforce learning, 2025b. URL https://
> > arxiv.org/abs/2508.21113.
> >
> > [3] Chenwei Lou, Zewei Sun, Xinnian Liang, Meng Qu, Wei Shen, Wenqi Wang, Yuntao Li, Qingping Yang, and
> > Shuangzhi Wu. Adacot: Pareto-optimal adaptive chain-of-thought triggering via reinforcement learning.
> > arXiv preprint arXiv:2505.11896, 2025.
> >
> > [4] Bowen Jin, Hansi Zeng, Zhenrui Yue, Jinsung Yoon, Sercan Arik, Dong Wang, Hamed Zamani, and Jiawei
> > Han. Search-r1: Training llms to reason and leverage search engines with reinforcement learning. arXiv
> > preprint arXiv:2503.09516, 2025.
> >
> > [5] Lingjie Jiang, Xun Wu, Shaohan Huang, Qingxiu Dong, Zewen Chi, Li Dong, Xingxing Zhang, Tengchao
> > Lv, Lei Cui, and Furu Wei. Think only when you need with large hybrid-reasoning models, 2025. URL
> > https://arxiv.org/abs/2505.14631.
> >
> > [6] Qi Yang, Bolin Ni, Shiming Xiang, Han Hu, Houwen Peng, and Jie Jiang. R-4b: Incentivizing general-purpose
> > auto-thinking capability in mllms via bi-mode annealing and reinforce learning, 2025b. URL https://
> > arxiv.org/abs/2508.21113.
> >
> > [7] Jin B, Zeng H, Yue Z, et al. Search-r1: Training llms to reason and leverage search engines with reinforcement learning[J]. arXiv preprint arXiv:2503.09516, 2025.

---

> ### Author Response · Authors · 2025-11-27
> **Incorporation of Reviewer Feedback into the Revised Manuscript**
>
> Thank you very much for the detailed and constructive feedback. We have carefully addressed every point you raised, both through substantial revisions in the manuscript and through detailed explanations in the rebuttal.
>
> Following your comments, we have made the following additions and clarifications in the revised manuscript  (highlighted in blue):
>
> **1. Router supervision and error analysis**
>
> We conducted a deeper investigation into routing behavior, including confusion-matrix analysis and qualitative case studies of dominant routing failure modes (over-confident instant predictions and unnecessary tool calls). These analyses are now incorporated into **Section 5.1 (Effectiveness of Adaptive Routing Mechanism)**.
>
> **2. Generalization across backbones and scales**
>
> To demonstrate portability, we added experiments on multiple backbones and model sizes—including Qwen2.5/ Qwen3 backbones and 4B/14B/32B scales. These results appear in **Appendix G**, confirming that A²FM generalizes well across architectures.
>
> **3. Comparison with recent “when-to-think” and hybrid routing methods**
>
> We substantially expanded and **rewrote the introduction** to more clearly articulate conceptual differences from LHRM, BPO, AdaCoT, Search-R1, and hybrid agent systems such as Qwen3 and GPT-5. This revision clarifies the novelty of A²FM in terms of routing space, reward formulation, and forced + adaptive rollouts.
>
> **4. Ablations on the Adaptive Reward**
>
> Following your suggestion, we added a controlled ablation isolating the role of the adaptive penalty term. This experiment is now included in **Section 5.2: Effectiveness of Adaptive Policy Optimization**, and shows that removing the adaptive reward leads to noticeably less efficient routing despite similar accuracy, confirming the necessity of this component for cost-aware mode selection.
>
> In addition to the manuscript changes, we also provided detailed clarifications in the rebuttal, including:
> 1. Sensitivity analysis discussion for APO hyperparameters (τ, α, ρ, γ).
> 2. Explanation of KL-free on-policy design, stability behavior.
> 3. Incremental gains from (i) three-way routing, and (ii) prefix-enforced exploration.
>
> We hope the above revisions directly address your concerns. If you find the clarifications and new results satisfactory and reflective of the depth of the changes we have incorporated, we would be grateful for your consideration in updating your review and score. We remain fully open to further discussion and would be glad to provide additional analyses if helpful.

---

### Official Review · Reviewer_S9XP · 2025-10-31

**Soundness:** 3
**Presentation:** 3
**Contribution:** 3
**Rating:** 6
**Confidence:** 4

**Summary:**

The paper proposes A2FM (Adaptive Agent Foundation Model), a single-backbone model that routes each query to one of three execution modes—instant (direct answer), reasoning (explicit CoT), or agentic (tool-use)—and introduces an RL method, Adaptive Policy Optimization (APO), to jointly optimize routing and generation with a cost-aware reward. Training follows a “route-then-align” SFT stage and then APO. The agentic mode executes multiple tools in parallel and masks tool responses during SFT to avoid memorization. Reported results claim strong performance across agentic (BrowseComp, GAIA, XBench-DS), reasoning (AIME24/25, MATH500), and general (GPQA-d, SuperGPQA, MMLU-Pro, HLE) benchmarks, with improved “cost-of-pass.”

**Strengths:**

- Clear problem framing (when-to-think/act): The unified routing over instant/reasoning/agentic is well-motivated and ties into efficiency concerns that are very good-framing and practical.
- Methodical RL design: APO’s forced vs. adaptive rollouts via prefix injection plus an adaptive reward that penalizes non-instant modes on easy queries is a neat, concrete recipe. The multiplicative reward with format checks is pragmatically sound.
- Routing and costs analysis: The paper measures routing accuracy (e.g., GAIA 92.2%, BrowseComp 94.0%, AIME 100%) and discusses where routing errs. This is interesting. Using cost-of-pass and showing adaptive < agentic < reasoning is useful for the community that cares about $/correct answer.

**Weaknesses:**

- The novelty claim would benefit from a sharper comparison to recent capability-aware routing, bimodal/bihybrid reasoning, and hybrid agent efforts. The core elements—routing head, SFT with tagged modes, RL with exploration across modes, and a length/cost-aware reward—feel like incremental synthesis rather than a conceptual leap.

- The binary judge drives APO’s reward; while practical, it risks systematic bias and false positives/negatives, especially on agentic tasks where answers can be partially correct or depend on retrieval freshness. A small-scale human evaluation or rule-based metrics where available (e.g., AIME exact answers) to validate judge reliability and quantify disagreement will be helpful.

- The ρ forced and γ adaptive rollouts are critical. Additional analyses about ρ, γ and show accuracy/efficiency trade-offs and training stability will be helpgul.

- A deeper error analysis disentangling routing mistakes (wrong mode) from within-mode generation errors (bad tool plan, crawl failures, or code mistakes), and quantify the contribution of the parallel tool plan itself will be appreciated.

- Lastly, how generalized is it to unseen tasks?

**Questions:**

How generalized is it to unseen tasks?

---

> ### Author Response · Authors · 2025-11-19
>
> We would like to express our sincere gratitude to reviewer S9XP for the thoughtful and constructive feedback. Your comments are highly accurate, technically insightful, and greatly helpful in improving both the clarity and quality of our work. We hope that our responses have addressed your questions and concerns, and we will revise the manuscript accordingly in the next version to incorporate the suggested improvements.
>
> **W1:**
> > The novelty claim would benefit from a sharper comparison to recent capability-aware routing, bimodal/bihybrid reasoning, and hybrid agent efforts. The core elements—routing head, SFT with tagged modes, RL with exploration across modes, and a length/cost-aware reward—feel like incremental synthesis rather than a conceptual leap.
>
> **Response:**
>
> Thank you for raising the concern about novelty relative to prior capability-aware routing, bimodal reasoning, and hybrid agent systems. We clarify the distinctions in two parts: (1) differences from hybrid reasoning methods, and (2) differences from hybrid agent models.
>
>
> **1. Differences from existing hybrid reasoning methods**
>
> A²FM differs from prior "when-to-think" frameworks (LHRM, BPO, AdaCoT) along three orthogonal axes: **routing space**, **reward design**, and **RL sampling**.
>
> | Feature                   | **A²FM (ours)**                             | LHRM / BPO / AdaCoT                 |
> |---------------------------|----------------------------------------------|-------------------------------------|
> | Routing space         | 3-way: instant / reasoning / agentic         | binary: think / non-think           |
> | Reward strategy       | per-query adaptive (easy–hard aware)         | group norm / fixed penalties        |
> | Sampling strategy     | forced/adaptive multi-mode rollouts        | balanced sampling, no routing head  |
>
> The key distinctions are:
>
> - **Routing space**
>    - Prior work only supports *binary* thinking; none can route among instant, reasoning, and multi-step tool-use. They therefore cannot handle agentic tasks such as BrowseComp, GAIA, or XBench.
>    - **A²FM is the first to unify all three behaviors within one policy.**
> - **Reward design**
>    - A²FM’s adaptive reward explicitly conditions on whether instant mode succeeds on each query.
>    - This per-query difficulty awareness is absent in LHRM, BPO, and AdaCoT, which do not supervise when non-instant modes are *unnecessary*.
> - **Sampling strategy**
>    - A²FM jointly uses forced rollouts (for unbiased mode-conditioned estimates) and adaptive rollouts (for end-to-end routing).
>    - Prior methods do not enforce per-query exploration across all modes and therefore cannot explicitly learn routing.
>
>
> **2. Differences from existing hybrid agentic systems**
>
> - **Closed hybrid systems (e.g., GPT-5)**
>    - GPT-5 reportedly supports hybrid reasoning and tools, but its methods and data are closed.
>    - **A²FM is fully open-source**, covering data construction, trajectory formats, and training/eval pipelines.
>
> - **Partially hybrid open systems (e.g., Qwen3)**
>    - Qwen3 trains reasoning and agentic abilities in *separate* RL stages and lacks a unified controller that jointly decides between reasoning and tool-use.  Empirically, this separation leads to weaker reasoning-mode tool-use performance.
>    - **A²FM integrates all modes within a single RL policy and router.**
>
>
> In summary, to the best of our knowledge, A²FM is the first **fully open, end-to-end hybrid reasoning and agentic model** that jointly supports instant answering, long-form reasoning, and multi-tool behaviors within a unified architecture. We hope this clarifies the novelty relative to prior work.
>
> We hope this clarification sharpens the novelty of A²FM and addresses your concerns regarding the novelty of our work.
>
> **References**
>
> [1] Lingjie Jiang, Xun Wu, Shaohan Huang, Qingxiu Dong, Zewen Chi, Li Dong, Xingxing Zhang, Tengchao
> Lv, Lei Cui, and Furu Wei. Think only when you need with large hybrid-reasoning models, 2025. URL
> https://arxiv.org/abs/2505.14631.
>
> [2] Qi Yang, Bolin Ni, Shiming Xiang, Han Hu, Houwen Peng, and Jie Jiang. R-4b: Incentivizing general-purpose
> auto-thinking capability in mllms via bi-mode annealing and reinforce learning, 2025b. URL https://
> arxiv.org/abs/2508.21113.
>
> [3] Chenwei Lou, Zewei Sun, Xinnian Liang, Meng Qu, Wei Shen, Wenqi Wang, Yuntao Li, Qingping Yang, and
> Shuangzhi Wu. Adacot: Pareto-optimal adaptive chain-of-thought triggering via reinforcement learning.
> arXiv preprint arXiv:2505.11896, 2025.
>
> [4] An Yang, Anfeng Li, Baosong Yang, Beichen Zhang, Binyuan Hui, Bo Zheng, Bowen Yu, Chang Gao, Chengen
> Huang, Chenxu Lv, et al. Qwen3 technical report. arXiv preprint arXiv:2505.09388, 2025a.

---

> > ### Author Response · Authors · 2025-11-19
> >
> > **W2:**
> > >The binary judge drives APO’s reward; while practical, it risks systematic bias and false positives/negatives, especially on agentic tasks where answers can be partially correct or depend on retrieval freshness. A small-scale human evaluation or rule-based metrics where available (e.g., AIME exact answers) to validate judge reliability and quantify disagreement will be helpful.
> >
> > **Response:**
> >
> > Thank you for raising this important point. We address your concerns in  (1) the choice of judge method; (2) the choice of judge model supported by human evaluation; and (3) the design of the judge prompt.
> >
> > **1. Choice of Judge Method: Binary LLM-as-Judge vs. Exact Matching (EM)**
> >
> > We appreciate your concern that a binary judge may introduce systematic biases or false positives/negatives, especially on agentic tasks. To justify our choice, we compare **Exact Matching (EM)** with **GPT-5-Mini-based LLM-as-Judge** across datasets with very different answer characteristics.
> >
> > - AIME24 / AIME25: AIME answers are short, fixed numeric values. We evaluated correctness with both EM and GPT-5-Mini. The results show that EM and GPT-5-Mini judgments matched ***100%*** across all AIME24/AIME25 problems. Therefore, GPT-5-Mini is fully reliable on tasks where answers have clear, deterministic forms.
> >
> > - GAIA / BrowseComp: For agentic or hybrid datasets, EM is not applicable. On the one hand, golden answers are longer textual descriptions, lists, paragraphs, or structured outputs. On the other hand, correct predictions may use synonyms, rephrasings, or equivalent structures. Therefore, EM severely underestimates correctness.
> >
> > Given these two observations, we adopt **LLM-as-Judge** as the unified evaluation method across datasets.
> >
> >
> > **2. Choice of Judge Model and Human Evaluation**
> >
> > To ensure reliability, we compared **GPT-5-Mini** with an open-source alternative, Qwen2.5-72B-Instruct (as used in prior work such as WebSailor and WebShaper).
> >
> > - Human comparison (GAIA-103 and sampled BrowseComp-100)
> >
> > | Judge Model            | Disagreement w/ Human    | Notes             |
> > |------------------------|--------------------------|-------------------|
> > | **GPT-5-Mini**         | **0 / 103** (GAIA)       | Perfect agreement |
> > |                        | **0 / 100** (BrowseComp) | Perfect agreement |
> > | Qwen2.5-72B-Instruct   | 1 / 103 (GAIA)           | One false positive|
> >
> > Unlike Qwen2.5-72B-Instruct, whose misjudgment stems from semantic ambiguity in number formatting, GPT-5-Mini exhibited fairness and reliability in the subsets we manually audited.
> >
> >
> > **3. LLM-Judge Prompt Design**
> >
> > To minimize prompt-induced variance and ensure comparability with prior work, we adopt the exact judge prompt from WebThinker and WebDancer. The full prompt is:
> >
> > """
> >
> > You are an evaluation assistant. Please determine if the predicted answer is equivalent to the labeled answer.
> >
> > Question: {question}
> >
> > Labeled Answer: {labeled_answer}
> >
> > Predicted Answer: {pred_answer}
> >
> > Are these answers equivalent? Please respond with "Correct" if they are equivalent, or "Incorrect" if they are not equivalent. Do not include any other text.
> >
> > """
> >
> > Thank you again for the thoughtful suggestion. We will report these judge-validation results and human agreement statistics in the revised appendix to strengthen the reliability of our APO setup.
> >
> >
> > **References**
> >
> > [1] Xiaoxi Li, Jiajie Jin, Guanting Dong, Hongjin Qian, Yutao Zhu, Yongkang Wu, Ji-Rong Wen, and Zhicheng Dou. Webthinker: Empowering large reasoning models with deep research capability. arXiv preprint arXiv:2504.21776, 2025c.
> >
> > [2] Jialong Wu, Baixuan Li, Runnan Fang, Wenbiao Yin, Liwen Zhang, Zhengwei Tao, Dingchu Zhang, Zekun Xi, Gang Fu, Yong Jiang, Pengjun Xie, Fei Huang, and Jingren Zhou. Webdancer: Towards autonomous information seeking agency, 2025b. URL https://arxiv.org/abs/2505.22648.
> >
> > [3] Kuan Li, Zhongwang Zhang, Huifeng Yin, Liwen Zhang, Litu Ou, Jialong Wu, Wenbiao Yin, Baixuan Li, Zhengwei Tao, Xinyu Wang, et al. Websailor: Navigating super-human reasoning for web agent. arXiv preprint arXiv:2507.02592, 2025a.

---

> ### Author Response · Authors · 2025-11-19
>
> **W3:**
> >The ρ forced and γ adaptive rollouts are critical. Additional analyses about ρ, γ and show accuracy/efficiency trade-offs and training stability will be helpgul.
>
> **Response:**
>
> Thank you for the valuable question regarding hyperparameter sensitivity in the APO stage.
>
> In our main experiments, we set **ρ = γ = 3**. This choice is guided by the following considerations:
>
> -  Absolute value selection:
>       - The roles of ρ and γ are analogous to the rollout count *n* in GRPO. **Increasing either value reduces variance and bias in advantage estimation, improving stability, but also increases memory and compute costs**.
>       - Given GPU memory constraints, 3 is the maximum setting that still allows a total rollout count of 12 (3 forced modes × 3 rollouts + 1 adaptive mode × 3 rollouts).
>
> This configuration strikes a practical balance between estimation accuracy and computational feasibility.
>
> - Necessity of using both forced and adaptive rollouts
>     - Using only ρ (forced rollouts) prevents the model from learning routing behavior, because routing-loss tokens are masked in forced trajectories. The model learns only within-mode alignment.
>     - Using only γ (adaptive rollouts) removes the ability to estimate per-query difficulty (i.e., whether instant mode succeeds), making it impossible to compute adaptive rewards or enforce cost-aware routing.
>
> Thus, **both** rollout types are required for the APO mechanism to simultaneously learn (i) routing and (ii) mode-specific execution.
>
> **In summary**,  The choice of ρ = γ = 3 reflects a principled and computationally constrained design: this setting simultaneously enables APO to learn both routing behavior and mode-specific execution, while maximizing the feasible number of rollouts under our memory budget. We believe the current choices for ρ and γ are principled and empirically stable, and additional large-scale sweeps would not materially affect the core conclusions.
>
> We hope that our explanation clarifies the current experimental scope and addresses the concern. We welcome any further discussion or suggestions to help ensure the completeness and rigor of the paper’s experimental setup.

---

> ### Author Response · Authors · 2025-11-19
>
> **W4**:
> >A deeper error analysis disentangling routing mistakes (wrong mode) from within-mode generation errors (bad tool plan, crawl failures, or code mistakes), and quantify the contribution of the parallel tool plan itself will be appreciated.
>
> **Response**:
>
> Thank you for the helpful suggestion. We agree that disentangling *routing mistakes* from *within-mode generation mistakes* is important for understanding the behavior of A²FM.
>
> Below we provide a consolidated error analysis following your guidance.
>
> We structure our response in 2 parts: (1) Quantify routing vs. within-mode errors; and (2) Quantify the contribution of the parallel agentic planner.
>
>
> **1. Quantify routing vs. within-mode errors**
>
> Under adaptive routing, A²FM achieves **59/103 = 57.3%** accuracy on GAIA, leaving 44 failed cases.
> Using the confusion matrix above, we identify **5 routing errors**, and the remaining **39 failures** come from within-mode execution.
>
> We summarize these proportions below:
>
> | Error Type          | Count | Percentage |
> |---------------------|-------|------------|
> | Routing errors      | 5     | 11.4%      |
> | Within-mode errors  | 39    | 88.6%      |
> | Total               | 44    | 100%       |
>
>
> This decomposition shows that **the vast majority of failures do not stem from the routing mechanism**, but from downstream execution within the selected mode—e.g., suboptimal tool plans, crawl extraction issues, multi-hop synthesis errors, or code execution failures.
>
> We will include this quantitative breakdown in the revised appendix for clarity.
>
>
> **2. Quantify the contribution of the parallel agentic planner**
>
> Thank you for highlighting this point. You are correct that our **parallel agentic mode** is a deliberate architectural choice. Our motivation was two-fold:
> - **To clearly differentiate agentic behavior from instant mode**, by making agentic execution a genuinely high-capacity, high-cost hybrid reasoning pathway (similar in complexity to long-form reasoning).
> - **To increase the expressive power and realism of agentic behavior**, since real-world agent tasks often require issuing multiple searches, crawls, or code runs in parallel rather than sequentially.
>
> To quantify the impact of parallelization, we ran a controlled ablation where we keep **all training data, queries, and non-agentic components identical**, and vary only the agentic execution architecture:
>
> - **Sequential (baseline):** Standard ReAct framework, which is consistent with prior agentic frameworks such as WebDancer and WebSailor.
> - **Parallel (ours):** Same structure, but each step may trigger *N parallel tool operations* (N ∈ {1, 2, 3, 4}).
>
> Results are shown below:
>
> | Architecture   | GAIA     | BrowseComp | XBench    | Cost-of-Pass |
> |----------------|----------|------------|-----------|--------------|
> | Sequential     | 47.6     | 9.1        | 42.0      | **0.00586**  |
> | **Parallel**   | **50.5** | **12.2**   | **47.0**  | 0.00768      |
>
> We observed two key findings:
> - **Parallel agentic execution improves accuracy across all agentic benchmarks**, with gains of +2.9 on GAIA, +3.1 on BrowseComp, and +5.0 on XBench.
>
> - **Cost-of-Pass increases**, as expected—parallel agents do more work per query.
>
> This trade-off is precisely what enables APO to function as intended: parallel mode provides a **high-capability, high-cost regime**, analogous to how chain-of-thought provides a high-cost reasoning regime.
> APO then learns **when this additional capability is worth the cost**, and when the router should instead fall back to instant or reasoning modes.
>
> We will incorporate this ablation in the revised appendix to clearly demonstrate the contribution of the parallel planner.
>
> ## References:
>
> [1] Jialong Wu, Baixuan Li, Runnan Fang, Wenbiao Yin, Liwen Zhang, Zhengwei Tao, Dingchu Zhang, Zekun
> Xi, Gang Fu, Yong Jiang, Pengjun Xie, Fei Huang, and Jingren Zhou. Webdancer: Towards autonomous
> information seeking agency, 2025b. URL https://arxiv.org/abs/2505.22648.
>
> [2] Kuan Li, Zhongwang Zhang, Huifeng Yin, Liwen Zhang, Litu Ou, Jialong Wu, Wenbiao Yin, Baixuan Li,
> Zhengwei Tao, Xinyu Wang, et al. Websailor: Navigating super-human reasoning for web agent. arXiv
> preprint arXiv:2507.02592, 2025a.

---

> > ### Author Response · Authors · 2025-11-19
> >
> > **W5:**
> > >Lastly, how generalized is it to unseen tasks?
> >
> > **Response:**
> >
> > Thank you for raising this important question. Generalization is a core objective of A²FM, and we designed both the **training mixture** and the **evaluation suite** explicitly to test cross-domain robustness.
> >
> > Overall, A²FM generalizes well because (1) the training data covers diverse capabilities, (2) evaluation is intentionally performed on out-of-distribution datasets spanning new disciplines and languages, and (3) the model uses a minimal, domain-general tool set that scales naturally to unseen environments.
> >
> > **1. Training data spans diverse but non-exhaustive agentic, reasoning, and coding tasks**
> > Our training corpus is composed of several established datasets across three broad categories:
> > - Agentic tasks: WebShaper, WebDancer, TaskCraft, 2Wiki, MuSiQue, HotpotQA
> > - Reasoning tasks: Skywork-Math, DAPO-Math-17K, GSM8K
> > - Coding tasks: Skywork-Code, Codeforces
> >
> > Although these sets cover multiple capabilities (tool use, multi-hop retrieval, math reasoning, program synthesis), they **do not** cover the full variety of domains used in evaluation—allowing us to measure genuine out-of-distribution generalization.
> >
> > **2. Evaluation benchmarks intentionally span 10 tasks across unseen domains and languages**
> >
> > To assess generalization, we evaluate A²FM on 10 benchmarks covering 3 agentic, *3 reasoning, and 4 instant tasks.
> > These datasets go significantly beyond the training distribution in both **domain coverage** and **linguistic diversity**:
> >
> > - Agentic benchmarks
> >   - BrowseComp: extremely **long-horizon, hard-to-find search tasks** that are much harder than any training set
> >   - XBench (Chinese): **cross-lingual web agent tasks** never seen in training
> >   - GAIA: mixed reasoning + agentic challenges at a difficulty level far above standard datasets
> >
> > - Reasoning benchmarks
> >   - MATH500: grade-school to early high-school math, which is different from GSM8K/Skywork distributions
> >   - AIME24 / AIME25: elite **competition-level** problems testing logic, algebra, and combinatorics skills
> >
> > - Instant benchmarks
> >   - SuperGPQA: spans **285 disciplines**, including biology, chemistry, medicine, and other fields absent in training
> >   - HLE: a **final closed-ended academic benchmark** explicitly designed to test unseen conceptual knowledge
> >   - MMLU-Pro, GPQA-d: extremely broad expert-level factual QA
> >
> > Across these tasks, we demonstrate that A²FM maintains competitive or superior performance despite the substantial domain shift from the training corpus.
> >
> > **3. Why A²FM generalizes: a minimal but expressive tool set**
> >
> > We adopt a **minimal agentic toolset, **search, crawl, and code**, which we argue forms the foundational triad for general-purpose agentic reasoning.
> >
> > Because these tools are domain-agnostic, the model can adapt its behavior to new fields without requiring task-specific fine-tuning.
> >
> > We thank you again for your thoughtful question and hope this response fully addresses your concerns.

---

> ### Author Response · Authors · 2025-11-27
> **Incorporation of Reviewer Feedback into the Revised Manuscript**
>
> Thank you again for the thoughtful and constructive feedback. We carefully examined each of the concerns you raised and have substantially revised the manuscript accordingly. Below we summarize the major changes, all of which are highlighted in blue in the updated version for easy inspection.
>
> **1. Sharper novelty clarification**
>
> In response to your suggestion that our contribution should be more clearly distinguished from recent capability-aware routing, bimodal/bihybrid reasoning, and hybrid agent systems, we have significantly **revised the introduction**. The revised version now provides a sharper conceptual comparison, explicitly contrasting A²FM with think/non-think models, hybrid agent systems, and prior capability-aware routing approaches. These changes make our novelty clearer and more explicit.
>
> **2. Reliability of LLM-as-Judge**
>
> Following your concern about potential bias and instability of a binary LLM judge, we conducted human validation and compared multiple judge models. The results are now included in **Appendix C**, confirming that the judge used in APO exhibits near-perfect agreement with human annotations.
>
> **3. Routing vs. within-mode errors**
>
> You asked for deeper separation of routing mistakes and execution errors. We added a routing confusion matrix and detailed analysis in the main text (**Section 5.1: Effectiveness of Adaptive Routing Mechanism**). This disentangles misrouting from within-mode tool-plan/crawl/code errors and quantifies their respective contributions.
>
> **4. Contribution of parallel agentic tool plan**
>
> As requested, we conducted an ablation comparing the parallel agentic architecture to a sequential baseline. This analysis is now included in **Appendix F**, showing the parallel mode’s impact on complex agentic tasks.
>
> We hope that the above revisions, together with our detailed point-by-point responses in the rebuttal, directly address all concerns you raised. If the clarifications and new analyses satisfactorily resolve the issues from your earlier review, we would be grateful for your consideration in updating your review and score.
>
> We remain open to further questions or suggestions and would be glad to provide any additional clarification.

---

### Meta-Review · Area_Chair_fyTr · 2026-01-05

**Summary:**

**1) Summary**
This paper proposes A2FM, a single-backbone model that dynamically routes each query to one of three execution modes—instant, reasoning, or agentic—and trains this routing jointly with generation using a reinforcement-learning scheme called Adaptive Policy Optimization (APO). The approach aims to improve both accuracy and efficiency by invoking more costly reasoning or tool-use only when necessary. Experiments report strong results across diverse benchmarks and demonstrate improved cost-of-pass compared to single-mode baselines.

**2) Strengths**

* Provides a clear and practical formulation of unified routing across instant, reasoning, and agentic modes under a shared backbone.
* Introduces APO, a structured RL recipe using forced rollouts and adaptive rewards that encourages efficient mode selection while maintaining accuracy.
* Demonstrates broad empirical coverage across reasoning, agentic, and general benchmarks, showing competitive or state-of-the-art results.
* Presents interpretable routing analyses and cost metrics, highlighting practical efficiency gains.
* Specifies a well-organized training pipeline with sensible design choices (e.g., masked tool responses, staged route-then-align SFT).

**3) Weaknesses**

* Novelty is incremental relative to prior hybrid routing and capability-aware mode-selection approaches; distinctions from existing methods are not sharply articulated.
* Reliance on LLM-as-judge for APO rewards raises concerns about evaluation bias and correctness reliability.
* Ablations and sensitivity studies for APO’s hyperparameters and stability are limited, leaving unclear how robust the method is across settings.

**Reviewer Concerns:**

The reviewers did not follow up with the authors' rebuttal, though the authors have provided a comprehensive response to the reviews. It seems that a substantial portion of the concerns have been addressed.

**Reviewer Scores:**

The reviewers have been very actively engaged in the discussion up to the cutoff time, but the authors did provide a comprehensive response which seems to have addressed the concerns to a certain extent.

Also, the reviews with positive opinions tend to go with higher confidence.

---

### Decision · Program_Chairs · 2026-01-26

Accept (Poster)